# Monocular Normal Estimation via Shading Sequence Estimation

**Zongrui Li[1]*   Xinhua Ma[1]*   Minghui Hu[1]   Yunqing Zhao[2]   Yingchen Yu[2]**

**Qian Zheng[3,4]   Chang Liu[5]†   Xudong Jiang[1]   Song Bai[2]**

[1]School of Electrical and Electronic Engineering, Nanyang Technological University
[2]ByteDance   [3]College of Computer Science and Technology, Zhejiang University
[4]The State Key Lab of Brain-Machine Intelligence, Zhejiang University
[5]MoE Key Laboratory of Interdisciplinary Research of Computation and Economics,
Shanghai University of Finance and Economics
https://github.com/LMozart/ICLR2026-RoSE.git

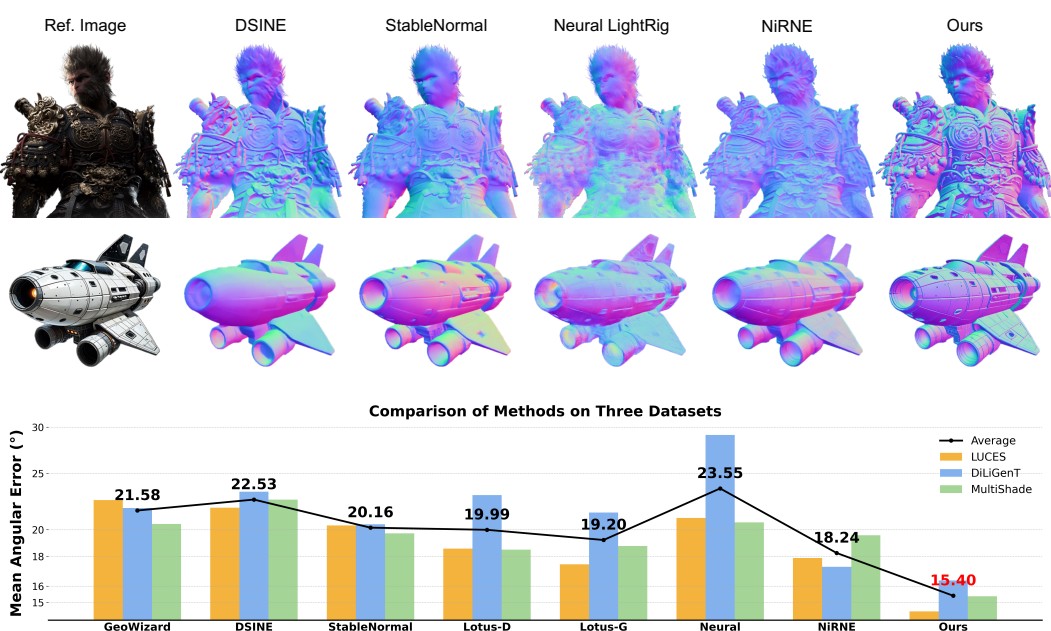

Figure 1: We present **RoSE**, a method using a video generative model for monocular normal map estimation, built on a new paradigm that reformulates normal estimation as a shading sequence estimation task. Results on complex and diverse scenarios show that RoSE reconstructs fine-grained geometric details and generalizes robustly to unseen datasets, achieving state-of-the-art performance in object-based monocular normal estimation on benchmark datasets.

## Abstract

Monocular normal estimation aims to estimate the normal map from a single RGB image of an object under arbitrary lights. Existing methods rely on deep models to directly predict normal maps. However, they often suffer from *3D misalignment*: while the estimated normal maps may appear to have a correct appearance, the reconstructed surfaces often fail to align with the 3D geometry. We argue that this misalignment stems from the current paradigm: the model struggles to distinguish and estimate varying geometry represented in normal maps, as the differences in underlying geometry are reflected only through relatively subtle color variations. To address this issue, we propose a new paradigm that reformulates normal estimation as shading sequence estimation, where shading sequences are more sensitive to various geometry information. By learning to infer the shading

*Equal contribution.
†Corresponding author.

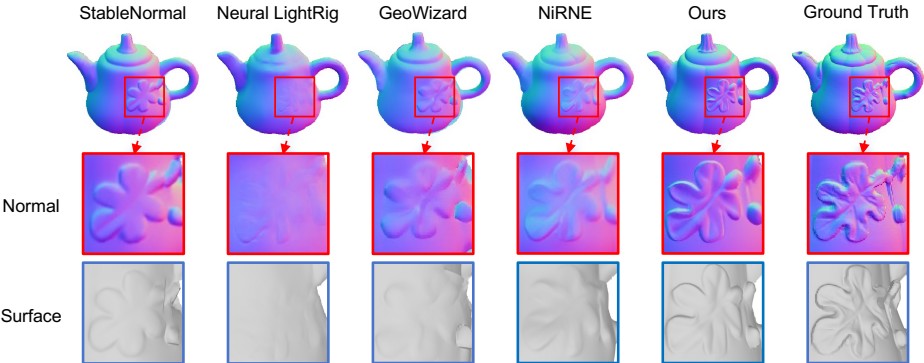

Figure 2: **Illustration of 3D misalignment.** The estimated normal maps of previous methods may appear to have an overall correct appearance, yet the reconstructed surfaces often fail to align with the accurate geometry details, showing over-smooth results. In contrast, our predicted normal maps exhibit better 3D alignment, leading to more faithful surface reconstruction.

sequence of an object, the model can better capture underlying 3D geometry and thereby produce more accurate normal predictions. Building on this paradigm, we present **RoSE**, a method that leverages image-to-video generative models to predict shading sequences, which are then converted into normal maps by solving a simple ordinary least-squares problem. To enhance robustness and better handle complex objects, RoSE is trained on a synthetic dataset, MultiShade, with diverse shapes, materials, and light conditions. Experiments demonstrate that RoSE achieves state-of-the-art performance on both synthetic and real-world benchmark datasets for object-based monocular normal estimation.

# 1 INTRODUCTION

Normal maps represent 3D geometry by specifying the surface orientation at each pixel point, making them essential for a wide range of applications, including relighting (Tiwari et al., 2024; Sang & Chandraker, 2020; Li et al., 2018; Liu et al., 2020; Li et al., 2023), 3D reconstruction (Ye et al., 2025), and rendering (Zeng et al., 2025; Pharr & Humphreys, 2015). In early methods, obtaining accurate normal maps required specialized hardware setups and was therefore costly for broad use. This motivates the development of techniques for monocular normal estimation, which aim to reliably infer normal maps directly from casually captured RGB images.

Specifically, to achieve accurate monocular normal estimation, prior works (Yoon et al., 2016; He et al., 2024b; Li et al., 2015; 2024b; Fu et al., 2024; Bae et al., 2021) directly predict normal maps from a single RGB image using deep neural models. Despite achieving promising results, these methods often produce normal maps that appear to have a correct appearance but fail to remain consistent with the underlying 3D geometry. We refer to this limitation as "**3D misalignment**" (see Fig. 2). We argue that this limitation arises from the current paradigm, where the model learns to recover geometry primarily by aligning with the color representation of normal maps. As a result, the deep model struggles to distinguish and estimate fine geometric details because normal maps represent geometry in a highly compact form, where geometry variations across different positions appear only as subtle color differences. This issue becomes particularly severe in monocular normal estimation, where geometric details in the input are inherently more ambiguous, thus difficult to recover.

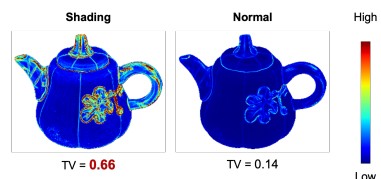

Figure 3: **Validation of sensitivity to geometry variations for different representations**, including the proposed shading sequence (left) and the normal map (right), measured by average total variation (TV). TV is computed as the mean magnitude of the first-order image's gradient in terms of different representations, where higher TV indicates stronger sensitivity to spatial geometric variation.

To reduce 3D misalignment, this paper proposes a new paradigm for normal estimation built on a redesigned training target. The main idea is to adopt a representation that is more sensitive to geometric variation as the ground-truth signal to be supervised, thereby enhancing the network's ability to distinguish and estimate geometric details. Guided by this intuition, we propose using a **shading sequence**, defined as the clamped dot product between the normal map and a set of canonical light directions, as the new training target. The idea of using a shading sequence is motivated by two key observations. First, shading sequences capture geometric variation through brightness variations, while excluding material influences, making them sensitive only to geometric variations, as illustrated in Fig. 3. Second, predicting the shading sequence given canonical light directions is equivalent to predicting the normal map, as the shading sequence can be losslessly converted to the normal map (Yu et al., 2010) using an Ordinary Least Squares (OLS) solver.

Based on the new paradigm, we propose **RoSE**, a method that **R**eformulating normal estimation as the **S**hading sequence **E**stimation based on the monocular input image. Specifically, as the shading sequence can be regarded as a video, we leverage recent image-to-video generative models (Voleti et al., 2024; Blattmann et al., 2023) to predict shading sequences. A key benefit of this design is that the rich lighting priors encoded in large-scale video models facilitate the generation of accurate shading variations. Once the shading sequence is obtained, we recover the normal map using an OLS solver (Woodham, 1980). In practice, to enhance robustness when handling objects with more complex materials and lights, we train our model on a collected dataset named MultiShade, enriched with more diverse materials and light conditions compared to previous datasets. Experimental results demonstrate that our method achieves superior performance compared to state-of-the-art methods. Overall, we summarize the main contributions of this paper as:

- We introduce a new paradigm that reformulates the task of monocular normal estimation as shading sequence estimation.
- Under the new paradigm, we propose RoSE, a monocular normal estimation method using an image-to-video generative model that predicts a shading sequence of an object under predefined canonical parallel lights and analytically derives normal maps from it.
- We train RoSE on MultiShade, a synthetic dataset with diverse material and light conditions. Experiments show that our method achieves state-of-the-art performance on multiple datasets, especially on the widely-used real-world benchmark datasets (*i.e*, DiLiGenT, LUCES).

## 2 RELATED WORKS

**Monocular Normal Estimation.** Despite persistent research efforts in monocular normal estimation, achieving high accuracy remains significantly challenging due to the complexity of this task, which requires the prediction of 3D geometry with highly limited input information. Early works (Eigen & Fergus, 2015; Do et al., 2020; Fouhey et al., 2013; Wang et al., 2015; Zhang et al., 2019; Bansal et al., 2016; Ladický et al., 2014; Li et al., 2015; Wang et al., 2020) relied on handcrafted features, empirical priors, or conventional deep neural networks. However, these methods often suffer from limited generalization ability. Recent methods based on generative models (Voleti et al., 2024; Fu et al., 2024; He et al., 2024a), physics-inspired deep networks (Bae & Davison, 2024), and auto-regressive frameworks (Ye et al., 2025) have demonstrated improved generalization ability and the capacity to estimate relatively accurate normal maps. However, the estimated normal maps suffer from 3D misalignment, a problem that stems from the current paradigm where the model fails to capture the compact information in the normal maps. To address this, other works (Tiwari et al., 2024; He et al., 2024b) attempt to first generate more input images under a set of predefined canonical lights and subsequently estimate normals from these multi-light images. Yet, the accuracy of such methods is often degraded by artifacts in the generated input images, resulting in more severe 3D misalignment. In contrast, we propose a new paradigm that uses shading sequences, a representation that is sensitive to geometry, as the training target, and leverage video generative models to estimate them based on the input image, which achieves improved 3D alignment.

**Video Generation.** Recent advances in video generation (Rombach et al., 2022; Peebles & Xie, 2023; Zhang et al., 2023; Ho et al., 2022) have significantly improved video synthesis quality. Methods such as (Blattmann et al., 2023; Deng et al., 2024; Kuaishou, 2024; Lin et al., 2024; Guo et al., 2023) can generate high-fidelity videos by enforcing temporal consistency across frames

through deep architectures such as temporal UNets (Blattmann et al., 2023; Guo et al., 2023) and Transformers (Deng et al., 2024; Lin et al., 2024). In 3D generation, video diffusion models are used to facilitate cross-view consistency (Voleti et al., 2024; Tang et al., 2024; Dai et al., 2023) to improve the quality of generated 3D models. In 3D estimation, recent work (Bin et al., 2025) employs a video diffusion model for normal estimation. They focus on predicting per-frame normals for an input video. In contrast, our work leverages the capability of video generative models to predict a shading sequence that follows a predefined light path consisting of multiple canonical parallel lights, using only a single input image. This enables accurate monocular normal estimation for objects with diverse shapes and materials.

**Shading Utilization.** In photometric stereo methods, shadings are often used to explain the behavior of a deep model in normal map estimation. Previous studies (Chen et al., 2020; Wei et al., 2025) show that learned features closely resemble shading sequences, motivating the use of shading sequence supervision to improve network performance (Wei et al., 2025). Inspired by these findings, we reformulate monocular normal estimation as a shading sequence estimation task and train a video diffusion model with shading sequences as targets.

## 3 METHODS

### 3.1 ON EQUIVALENCE OF NORMAL ESTIMATION AND SHADING SEQUENCE ESTIMATION

**Shading map and shading sequence**. In this paper, we define shading map (Wei et al., 2025) as:

$$\mathbf{S} \triangleq \{\mathbf{s}_\mathrm{p} = \max(\mathbf{n}_\mathrm{p} \cdot \mathbf{l}, 0) | \mathrm{p} \in \mathcal{P}\}, \tag{1}$$

where $\mathbf{n}$ is the normal map, and $\mathbf{l}$ is the direction of parallel light, $\max(., 0)$ is the nonlinear maximum operation that clamp the negative values, $\mathcal{P}$ is the points that belong to the object. Shading maps remove the effects of surface reflectance and occlusion-induced cast shadows while preserving the geometry information and the attached shadow. A sequence of shading maps obtained under multiple canonical lights, defined as a **shading sequence**, $\mathbf{S}^s \triangleq \{\mathbf{S}_i \mid i \in 1, \ldots, f\}$, provides sensitive cues to the underlying 3D geometry.

**Normal map estimation**. Given an observed image $\mathbf{I}$ of an object captured under arbitrary lights, the goal of monocular normal estimation is to recover the normal map $\mathbf{N} \triangleq \{\mathbf{n}_\mathrm{p} | \mathrm{p} \in \mathcal{P}\}$. This requires learning a mapping:

$$\Phi : \mathbf{I} \to \mathbf{N}. \tag{2}$$

Previous methods rely on deep models to learn a direct color mapping from a single RGB image to a normal map. This often produces a visually aligned appearance but inaccurate 3D geometry (*i.e*, the 3D misalignment). A more recent line of work (He et al., 2024b; Tiwari et al., 2024) first generates a series of RGB images under simple light sources and then estimates normals from them. The main idea of these works is to augment the input with additional generated images that provide additional cues, thereby improving the prediction of normal maps. However, as the materials, lights, and geometry in the input image become more complex, the process of generating additional RGB images itself introduces substantial bias, ultimately leading to more pronounced 3D misalignment artifacts.

**Shading sequence estimation**. The shading sequence under a set of predefined, non-coplanar, parallel lights (canonical lights) can be converted into a normal map. This enables us to shift the training target to shading sequence prediction with lights that vary along a predefined path, $\mathbf{L} \triangleq \{\mathbf{l}_i \mid i \in 1, \ldots, f\}$.

$$\Phi_S : \mathbf{I}_g \to \mathbf{S}^s, \tag{3}$$

where $\mathbf{I}_g$ denotes the grayscale input image. Then, the shading-to-normal estimation $\mathbf{S}^s \to \mathbf{N}$ can be solved via Ordinary Least Squares (OLS) (Woodham, 1980):

$$\mathbf{N} = \arg \min_{\mathbf{N}} \|\mathbf{N}^\top \mathbf{L} - \mathbf{S}^s\|^2 = (\mathbf{L}^\top \mathbf{L})^{-1} \mathbf{L}^\top \mathbf{S}^s. \tag{4}$$

The solution is unique when $\mathbf{L}$ is full rank. In practice, however, the $\max(\cdot, 0)$ operation causes a truncation effect, which makes the OLS estimate biased if OLS is applied directly to the shading sequence. To address this issue, we solve for the normal using only shadings with values greater than 0, treating them as valid equations in OLS.

Figure 5: **Pipeline of RoSE.** Given a monocular RGB image under arbitrary light, RoSE first converts it into a grayscale image, which is then used to generate the shading sequence via a video diffusion model. This generation is guided by two complementary feature representations extracted from a CLIP encoder and a VAE encoder. Finally, an ordinary least squares problem is solved analytically to estimate the normal map from the generated shading sequence. We train the video diffusion model while freezing the CLIP and the VAE encoder.

## 3.2 SHADING SEQUENCE-BASED TRAINING TARGET

Reformulating monocular normal estimation as shading sequence estimation introduces additional flexibility in designing the training target, since different choices of $\mathbf{L}$ yield different shading sequences. As long as each surface point is illuminated by at least three non-coplanar parallel light sources (*i.e*, the lighting matrix $\mathbf{L}$ is full rank in Eq. (4)), normal maps can be recovered from the shading sequence without information loss. In our setup, this means that each surface point should correspond to at least three positive shading values. In this paper, we adopt a classic ring light setup from photometric stereo (Zhou & Tan, 2010), where canonical lights are uniformly placed on a latitude ring in the upper hemisphere of the object's surface, each light oriented toward the surface center (see Fig. 4).

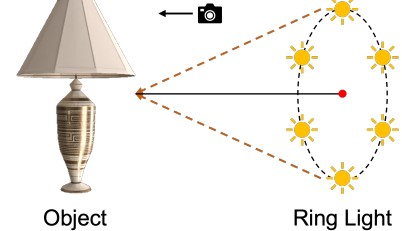

With an appropriate choice of the ring's latitude ($45°$ in our setup), these lights collectively illuminate all surface points. The remaining question is: *How many lights with distinct positions do we need, in particular the minimum number $l_{\min}$, to guarantee that every surface point is illuminated by at least three light sources with positive shading values?* We address this in Lemma 1.

**Lemma 1.** *Define a surface point as illuminated if its shading value is positive,* i.e, $\max(0, \mathbf{S}) > 0$. *Under a single parallel light, at least half of the upper hemisphere is illuminated. Therefore, $n = 2$ lights are sufficient to ensure that*

Figure 4: **Ring light setup**.

*every point on the hemisphere is illuminated once. To further guarantee that every point is illuminated by at least $m = 3$ different lights, the minimum number of lights is $l_{\min} = m \times n = 6$, with the light directions uniformly distributed along the ring.*

In our experiments, we observed that slightly increasing the number of light sources improved the accuracy of both normal and shading estimation. The best performance was achieved with 9 light sources, yielding a $0.74°$ improvement on LUCES Mecca et al. (2021) compared to 6 lights. While appropriately introducing additional light sources can further enhance accuracy, it also incurs longer training and convergence time as well as higher resource consumption. For instance, under the same settings, performance dropped by $1.31°$ under 12 lights.

## 3.3 ROSE: A MONOCULAR NORMAL ESTIMATOR BASED ON VIDEO GENERATIVE MODEL

**Architecture of RoSE** is shown in Fig. 5. Firstly, the shading generator $\mathbf{g}_\theta(\cdot)$ is designed to take grayscale images $\mathbf{I}_g$ as input, effectively eliminating redundant chromatic information that may distract the model from learning geometric cues. It produces grayscale shading sequences that follow a predefined light path, introducing structured patterns and temporal coherence well-suited to video generation models. In this paper, we implement $\mathbf{g}_\theta(\cdot)$ using a standard video diffusion U-Net composed of multiple spatial and temporal transformer blocks (Voleti et al., 2024; Blattmann et al., 2023). The grayscale input image $\mathbf{I}_g$ is used as an additional condition to guide the denoising process during shading sequence generation.

Specifically, following previous works (He et al., 2024b; Voleti et al., 2024; Blattmann et al., 2023), we adopt a similar dual-branch conditioning strategy that combines global guidance from CLIP embedding and local guidance from VAE latent concatenation to reuse the pre-trained weights of the model. 1) **CLIP embedding**: We extract a global feature vector $\mathbf{c}_g$ from the input image using a pretrained CLIP encoder. This semantic embedding is injected into the denoising U-Net via cross-attention, guiding the shading generation with object-level context. 2) **VAE latent concatenation**: to preserve spatial details, we encode the grayscale input $\mathbf{I}_G$ with a pretrained VAE encoder $\mathcal{E}$ and concatenate the resulting latent with the noisy latent $z_t$ at each denoising step. Since $\mathbf{I}_G$ is single-channel, we replicate it to three channels before feeding it into the VAE and CLIP encoders: $\mathbf{I}'_g = \text{repeat}(\mathbf{I}_g, B \times H \times W \rightarrow B \times H \times W \times 3)$, where $B$ is the batch size. By combining both conditioning techniques, the model better preserves the fine geometric structures of the input image during generation, which is particularly important for accurate normal estimation. Finally, the generated latents are decoded by the VAE decoder $\mathcal{D}$ and averaged across channels to obtain the final grayscale shading sequence.

**Training.** During training, we use the standard training objectives on latent space encoded by $\mathcal{E}$. The video generative model will learn to predict the noise given the noisy latent $\mathbf{z}_t$, $\mathbf{z}_0 = \mathcal{E}(\mathbf{S}^s)$, where $\mathbf{z}_t = \alpha_t \epsilon + \sigma_t \mathbf{z}_0$. The diffusion loss follows calculation of $\mathbf{z}_0$-reparameterization (Ho et al., 2020):

$$\mathcal{L}_{\text{diff}} = \mathbb{E}_{\mathbf{z}_0, c, t} \| \mathbf{z}_0 - \hat{\mathbf{z}}_0 \|^2, \hat{\mathbf{z}}_0 = \frac{\mathbf{z}_t - \alpha_t \mathbf{g}_\theta (\mathbf{z}_t | c', t)}{\sigma_t}. \tag{5}$$

where $\hat{\mathbf{z}}_0$ the one-step denoised version of $\mathbf{z}_t$.

**Dataset Curation**. To improve robustness when handling more complex materials and lights, we curate a dataset named MultiShade, featuring diverse shapes, materials, and light conditions to ensure robust generalization. MultiShade is built upon a list of pre-filtered 3D models (90K) curated from Objaverse (Deitke et al., 2023; He et al., 2024b), a widely adopted resource for 3D generation and reconstruction. For each object, we render observed images under three lighting setups: (1) **parallel lights** randomly placed around the object; (2) **point lights** with randomly sampled positions and intensities; and (3) **environment lights** using high-dynamic-range (HDR) maps sampled from a public collection of 780 real-world environments (Poly Haven, 2025). Each object is rendered from six distinct viewpoints (top, left, right, bottom, front, and one random view) to ensure comprehensive geometric coverage. To avoid lighting from the object's backside in each view, we apply view-dependent transformations to keep light sources in the upper hemisphere relative to the view direction. During rendering, we implement **material augmentation** to the dataset by either retaining the object's original texture or applying material augmentation. With a probability of $0.5$, an additional material is assigned from the MatSynth dataset (Vecchio & Deschaintre, 2024), which contains 5,657 high-quality PBR materials. More specifically, we assign a probability of $0.25$ to sample materials from the metallic category, while $0.25$ to extract materials from non-metallic categories such as plastic, wood, and fabric. This augmentation improves surface diversity and model robustness, especially for metallic materials. All images are rendered using Blender at a resolution of $576 \times 576$ following (Voleti et al., 2024), generating approximately 3 million image-normal pairs. Precomputed shading sequences under known canonical light sources are also provided. More details on rendering parameters, camera setup, and augmentation strategies are in the appendix.

## 4 EXPERIMENTS

### 4.1 EXPERIMENT SETUP

**Datasets**. We evaluate the proposed method on widely used benchmarks, including LUCES (Mecca et al., 2021) for near-light monocular normal estimation, DiLiGenT (Shi et al., 2016) for parallel-light settings, and a curated test set of 100 unseen objects from the Objaverse dataset (Deitke et al., 2023) rendered with diverse materials and light conditions.

**Baselines.** We compare RoSE with 7 other monocular normal estimation methods, *i.e*, GeoWizard (Fu et al., 2024), DSINE (Bae & Davison, 2024), StableNormal (Ye et al., 2024), Lotus-G & Lotus-D (He et al., 2024a), Neural LightRig (He et al., 2024b), and NiRNE (Ye et al., 2025).

**Implementation details and evaluation metrics.** All training experiments are conducted on $8 \times$ NVIDIA H100 GPUs with 80GB memory. The model is trained at a learning rate of $1 \times e^{-5}$, using

Table 1: Quantitative comparison in terms of MAE (↓) of the normal map on DiLiGenT benchmark dataset. Highlighted numbers indicate the best and second best results among monocular estimation methods.

| Method | BALL | BEAR | BUDDHA | CAT | COW | GOBLET | HARVEST | POT1 | POT2 | READING | Mean |
|---|---|---|---|---|---|---|---|---|---|---|---|
| GeoWizard | 16.85 | 14.58 | 26.38 | 21.82 | 19.54 | 17.70 | 29.78 | 21.86 | 19.97 | 29.42 | 21.79 |
| DSINE | 23.82 | 14.15 | 28.09 | 18.22 | 19.35 | 22.63 | 35.90 | 20.90 | 19.14 | 30.25 | 23.25 |
| StableNormal | 17.11 | 13.17 | 21.84 | 22.46 | 22.63 | 15.96 | 32.14 | 17.43 | 16.53 | 25.15 | 20.44 |
| Lotus-D | 36.83 | 11.29 | 21.68 | 23.93 | 22.62 | 13.93 | 34.99 | 21.45 | 17.14 | 25.49 | 22.94 |
| Lotus-G | 12.74 | 13.02 | 23.27 | 22.68 | 22.78 | 15.52 | 32.94 | 23.27 | 19.23 | 28.67 | 21.41 |
| Neural LightRig | 10.16 | 14.47 | 26.23 | 28.39 | 21.16 | 22.70 | 76.82 | 24.71 | 31.84 | 34.51 | 29.10 |
| NiRNE | 10.26 | 10.87 | 21.28 | 15.43 | 15.03 | 17.91 | 27.40 | 15.27 | 16.15 | 23.08 | 17.27 |
| **Ours** | **5.51** | **9.22** | **20.72** | **15.78** | **13.28** | **16.55** | **28.62** | **16.05** | **14.24** | **23.65** | **16.36** |

Table 2: Quantitative comparison in terms of MAE of the normal map on LUCES benchmark dataset (Mecca et al., 2021). Highlighted numbers indicate the best and second best results among monocular estimation methods.

| Method | BALL | BELL | BOWL | BUDDHA | BUNNY | CUP | DIE | HIPPO | HOUSE | JAR | OWL | QUEEN | SQUIRREL | TOOL | Mean |
|---|---|---|---|---|---|---|---|---|---|---|---|---|---|---|---|
| GeoWizard | 30.09 | 9.08 | 22.29 | 22.71 | 15.90 | 20.20 | 15.76 | 17.55 | 42.15 | 11.07 | 28.68 | 25.36 | 35.48 | 18.57 | 22.49 |
| DSINE | 26.88 | 15.00 | 9.53 | 22.34 | 15.82 | 22.65 | 32.02 | 14.42 | 36.95 | 16.26 | 27.46 | 23.76 | 25.26 | 17.19 | 21.82 |
| StableNormal | 9.58 | 9.36 | 31.39 | 20.80 | 14.73 | 29.40 | 11.88 | 20.80 | 37.55 | 8.25 | 23.23 | 21.10 | 27.24 | 19.49 | 20.34 |
| Lotus-D | 17.94 | 9.50 | 11.43 | 19.70 | 12.99 | 37.44 | 13.14 | 15.85 | 35.30 | 9.69 | 20.53 | 19.72 | 23.52 | 13.15 | 18.56 |
| Lotus-G | 17.82 | 8.66 | 10.89 | 19.71 | 12.90 | 23.26 | 12.59 | 16.94 | 35.32 | 10.69 | 18.94 | 20.65 | 24.05 | 11.74 | 17.44 |
| Neural LightRig | 9.52 | 11.95 | 21.71 | 20.66 | 15.25 | 18.08 | 25.13 | 18.54 | 39.67 | 19.78 | 23.40 | 23.35 | 25.32 | 20.97 | 20.95 |
| NiRNE | 10.55 | 12.00 | 17.35 | 20.62 | 16.14 | 15.78 | 12.57 | 15.85 | 34.99 | 10.37 | 22.46 | 22.41 | 21.90 | 17.34 | 17.88 |
| **Ours** | **9.09** | **5.94** | **6.84** | **17.58** | **12.70** | **13.80** | **8.26** | **14.14** | **36.79** | **5.93** | **19.60** | **19.99** | **21.34** | **10.66** | **14.48** |

AdamW as the optimizer. The diffusion architecture follows previous work (Voleti et al., 2024). More details can be found in the appendix. To assess the accuracy of predicted normal maps, following the common protocol in previous works (Ye et al., 2024; 2025; Bae & Davison, 2024; He et al., 2024a), we use the mean angular error (MAE) as the evaluation metrics for all experiments.

## 4.2 PERFORMANCE ON BENCHMARK DATASET

We conduct monocular normal estimation experiments on the DiLiGenT (Shi et al., 2016) and LUCES benchmark dataset (Mecca et al., 2021) to evaluate our method's ability in handling objects captured under distant and near-field light sources. For each object, we select 10 images with relatively centered lights so that the light can cover enough details. The index of the images used for testing can be found in the appendix.

**Quantitative analysis on normal estimation**. The results in Table 1 and Table 2 present the average MAE for each object across selected 10 images, and the average MAE across all objects. These quantitative results demonstrate a significant advantage of our method over the state-of-the-art method (16.36° for ours vs. 17.27° for NiRNE (Ye et al., 2025) on DiLiGenT dataset (Shi et al., 2016); 14.48° for ours vs. 17.44° for Lotus-G He et al. (2024a) on LUCES dataset (Mecca et al., 2021)). This validates the effectiveness of our method in achieving robust performance in normal estimation across various materials and light conditions. However, we observe that for certain objects, such as GOBLET in DiLiGenT and HOUSE in LUCES, our method does not rank within the top two. We attribute this to the model's inherent variance. Note that even the previous SOTA method, NiRNE, fails to deliver consistently strong performance across all cases. Another possible reason may be attributed to the training set used. We discuss this in Sec. 4.4.

**Qualitative analysis on normal estimation**. We present a qualitative comparison between our method and state-of-the-art methods in Fig. 6. Our method consistently recovers accurate object details in the estimated normal maps, achieving lower MAE. In contrast, previous methods often produce overly smooth results, distorted normal distributions, or significant artifacts (He et al., 2024b) (*e.g.*, the tail and back of the SQUIRREL). These results validate the effectiveness of the proposed shading sequence-based formulation and demonstrate RoSE's advantage in preserving fine shape details for accurate normal estimation.

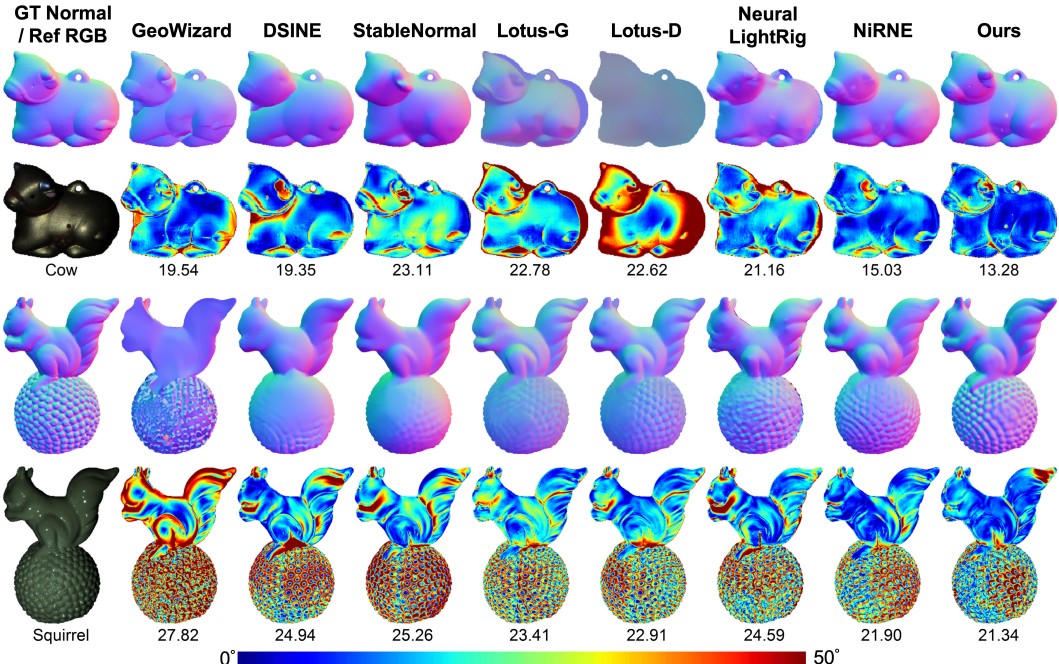

Figure 6: **Qualitative comparison** on selected objects from two benchmark dataset (COW from DILIGENT (Shi et al., 2016) and SQUIRREL from LUCES (Mecca et al., 2021). Row 1 & 3: normal map comparison. Row 2 & 4: error map comparison.) Best viewed in color with zooming in.

Table 3: Quantitative comparison in terms of **Mean** and **Median** Angular Errors of the normal map on MultiShade test set, and the percentage of objects below a specific error bound. Highlighted numbers indicate the best and second best results among monocular estimation methods.

| Method | Mean ↓ | Median ↓ | 3° ↑ | 5° ↑ | 7.5° ↑ | 11.25° ↑ | 22.5° ↑ | 30° ↑ |
|---|---|---|---|---|---|---|---|---|
| GeoWizard | 20.46 | 11.61 | 12.84 | 25.41 | 37.34 | 49.09 | 68.53 | 76.29 |
| DSINE | 22.53 | 14.04 | 12.38 | 22.47 | 32.18 | 43.27 | 65.19 | 74.16 |
| StableNormal | 19.71 | 11.23 | 6.83 | 18.67 | 34.65 | 50.08 | 71.66 | 79.48 |
| Lotus-D | 18.48 | 10.63 | 14.51 | 26.34 | 38.78 | 51.78 | 72.47 | 79.82 |
| Lotus-G | 18.76 | 10.65 | 14.67 | 27.13 | 39.19 | 51.63 | 71.83 | 79.54 |
| Neural LightRig | 20.59 | 11.36 | 17.65 | 27.59 | 37.90 | 49.69 | 70.85 | 78.54 |
| NiRNE | 19.57 | 13.57 | 4.06 | 11.92 | 25.53 | 42.10 | 71.42 | 81.21 |
| **Ours** | **15.37** | **7.78** | **26.99** | **38.38** | **49.00** | **60.32** | **78.30** | **84.28** |

**Analysis on shading sequence estimation.** We conduct quantitative analyses[1] of the predicted shading sequences on the LUCES dataset to illustrate RoSE's ability to recover accurate shading sequences. The shading map of all other methods (including the ground truth) is computed as the dot product between the lights' directions and the surface normal, with negative values clamped to zero. We use PSNR (↑), SSIM (↑), and LPIPS (↓) as the evaluation metrics, as shown in Table 4. The results demonstrate that RoSE achieves SOTA performance in predicted shading sequence, which also align with the results of normal estimation.

## 4.3 PERFORMANCE ON MULTISHADE

To further evaluate our method's performance across various lighting conditions and materials, the test set of the applied synthetic dataset consists of 100 unseen objects from Objaverse (Deitke et al., 2023). Each object is rendered with random materials selected from the MatSynth test set (Vecchio & Deschaintre, 2024). For lighting conditions, we employ one random point light, one directional (parallel) light, and two environmental lights selected from Poly Haven (Poly Haven, 2025) that are different from the training dataset. Each object is rendered from seven viewpoints, including the front, back, left, right, and top views, as well as two randomly sampled views. This setup yields a total of

---

[1]Please refer to the appendix for qualitative analyses.

Table 4: Quantitative comparison on estimated shading sequence in terms of PSNR (↑), SSIM (↑), and LPIPS (↓) on LUCES benchmark dataset (Mecca et al., 2021). Highlighted numbers indicate the best and second best results.

| Metrics | GeoWizard | DSINE | StableNormal | Lotus-D | Lotus-G | Neural LightRig | NiRNE | Ours |
|---|---|---|---|---|---|---|---|---|
| PSNR (↑) | 16.86 | 17.05 | 18.40 | 18.80 | 19.19 | 17.88 | 18.99 | **20.74** |
| SSIM (↑) | 0.6920 | 0.7199 | 0.7411 | 0.7492 | 0.7589 | 0.7139 | 0.7503 | **0.7744** |
| LPIPS (↓) | 0.2806 | 0.3100 | 0.2972 | 0.2868 | 0.2724 | 0.2831 | 0.2688 | **0.2583** |

Table 5: Quantitative analysis in terms of MAE and SNE of the normal map on LUCES benchmark dataset (Mecca et al., 2021). Highlighted numbers indicate the best and second best results.

| | GeoWizard | DSINE | StableNormal | Lotus-D | Lotus-G | Neural LightRig | NiRNE | Ours |
|---|---|---|---|---|---|---|---|---|
| MAE (↓) | 22.49 | 21.82 | 20.34 | 18.56 | 17.44 | 20.95 | 17.88 | **14.48** |
| SNE (↓) | 37.76 | 33.08 | 29.20 | 33.20 | 29.85 | 32.77 | 26.78 | **26.74** |

2800 test samples. Following the evaluation protocol in prior work (He et al., 2024b), we report the mean and median angular error (MAE) across all objects, as well as the percentage of objects with MAE below specified angular thresholds. As shown in Table 3, our method consistently outperforms baseline approaches across all metrics, with particularly strong performance under tighter thresholds (*i.e*, 3°-7.5°), highlighting the robustness and accuracy of the proposed RoSE.

## 4.4 ABLATION STUDY

We conduct ablation experiments using the LUCES benchmark dataset (Mecca et al., 2021) as the test set to analyze the effectiveness of the proposed RoSE and MultiShade. Additional experiments, analysis, and discussion are in the appendix.

**Validation on details alignment.** Following (Ye et al., 2025), we evaluate detailed alignment by computing the sharp normal error (SNE), *i.e*, the normal estimation error measured in boundary regions, on the estimated normal maps from the LUCES dataset, see Table 5. Our method achieves performance comparable to the state of the art method (NiRNE Ye et al. (2025)) and shows a clear advantage over other baselines. It is worth noting that NiRNE was trained on a dataset nearly $10\times$ larger and containing significantly more diverse and complex 3D models than ours. These results highlight that the proposed RoSE is capable of generating fine-grained details even with substantially lower resource consumption during training.

**Validation on negative-clamping on shading sequence.** After clamping negative values, the shading sequence is rescaled to the range $[-1, 1]$ (by applying a linear transformation $\mathbf{S} \mapsto \mathbf{S} \times 2 - 1$) to match the input requirements of the VAE encoder (Blattmann et al., 2023). This rescaling makes the shading sequence more sensitive to geometric variations. The effectiveness of this strategy is validated in Table 6 with comparison between 'ours' and 'ours w/o clamp'.

**Validation on material augmentation**. To evaluate the effectiveness of the proposed dataset, we train RoSE on the publicly available LightProp (He et al., 2024b) dataset as a reference. The key difference is that our dataset introduces additional material augmentation to increase material diversity. As shown in Table 6, training without material augmentation (w/o MA, using only the original object materials) yields slightly worse performance on the LUCES (Mecca et al., 2021) benchmark, while enabling material augmentation leads to notable improvements. These results demonstrate the effectiveness of the material augmentation.

**Validation on dataset impact**. We retrain the previous SOTA method on LUCES (*i.e*, Lotus-G) using our dataset ("Lotus-G+M"), resulting in consistent improvements and further validating the effectiveness of the proposed dataset. More importantly, when trained on the same dataset, our method still outperforms the baselines: Neural LightRig vs. "Ours+L" and "Lotus-G+M" vs. Ours, clearly demonstrating SOTA performance and highlighting its efficiency and competitiveness. Finally, we also observed that for some specific objects, such as HOUSE, retraining Lotus-G with our dataset resulted in decreased performance (35.32° for Lotus-G vs. 38.90° for "Lotus-G+M"). This suggests that dataset variations may affect estimation accuracy on certain objects.

Table 6: Ablation study in terms of MAE of the normal map on LUCES benchmark dataset (Mecca et al., 2021). In particular, "+M"("+L") means training on Multishade (LightProp) dataset, 'w/o clamp' means removing clamping on shading sequence. 'w/o MA' means training on dataset without material augmentation. Highlighted numbers indicate the best and second best results.

| Method | BALL | BELL | BOWL | BUDDHA | BUNNY | CUP | DIE | HIPPO | HOUSE | JAR | OWL | QUEEN | SQUIRREL | TOOL | Mean |
|---|---|---|---|---|---|---|---|---|---|---|---|---|---|---|---|
| Lotus-G | 17.82 | 8.66 | 10.89 | 19.71 | 12.90 | 23.26 | 12.59 | 16.94 | 35.32 | 10.69 | 18.94 | 20.65 | 24.05 | 11.74 | 17.44 |
| Neural LightRig | 9.52 | 11.95 | 21.71 | 20.66 | 15.25 | 18.08 | 25.13 | 18.54 | 39.67 | 19.78 | 23.40 | 23.35 | 25.32 | 20.97 | 20.95 |
| Ours w/o clamp | 11.63 | 7.75 | 12.66 | 17.79 | 12.32 | 15.72 | 9.30 | 14.08 | 40.77 | 4.64 | 20.65 | 21.18 | 21.01 | 12.84 | 15.88 |
| Ours w/o MA | 12.97 | 6.92 | 9.25 | 19.00 | 14.90 | 15.96 | 10.65 | 15.18 | 39.35 | 6.64 | 20.49 | 20.65 | 21.50 | 13.23 | 16.19 |
| Lotus-G+M | 16.21 | 8.95 | 7.11 | 16.57 | 11.50 | 22.41 | 16.40 | 14.04 | 38.90 | 13.14 | 24.69 | 19.78 | 18.96 | 14.35 | 17.36 |
| Ours+L | 9.37 | 7.03 | 8.46 | 19.42 | 12.24 | 14.05 | 8.73 | 15.06 | 38.14 | 5.81 | 20.94 | 20.22 | 22.19 | 11.23 | 15.21 |
| Ours w/ spiral | 16.32 | 9.25 | 10.97 | 20.23 | 16.78 | 16.70 | 12.17 | 16.06 | 39.68 | 7.38 | 22.25 | 21.86 | 22.50 | 14.29 | 17.60 |
| Ours w/ RGB | 10.36 | 8.56 | 8.99 | 18.28 | 13.89 | 11.56 | 9.22 | 13.70 | 38.49 | 5.74 | 19.80 | 20.45 | 20.78 | 13.99 | 15.27 |
| Ours w/ SVD XL | 8.69 | 7.68 | 9.16 | 18.34 | 12.43 | 12.15 | 8.47 | 14.11 | 37.60 | 6.94 | 18.73 | 19.10 | 19.28 | 11.38 | 14.58 |
| **Ours** | **9.09** | **5.94** | **6.84** | **17.58** | **12.70** | **13.80** | **8.26** | **14.14** | **36.79** | **5.93** | **19.60** | **19.99** | **21.34** | **10.66** | **14.48** |

**Validation on model variants impact**. We train RoSE using a different video diffusion backbone, namely Stable Video Diffusion XL (SVD XL) (Blattmann et al., 2023), on the MultiShade dataset. We denote this variant as "Ours w/ SVD XL". As shown in Table 6, this model achieves performance comparable to RoSE built on SV3D (14.58° for "Ours w/ SVD XL" and 14.48° for Ours). This demonstrates that our framework generalises well even when the backbone is pretrained on large-scale, general-purpose video data rather than a domain-specific object-centric dataset.

**Validation on gray-scale input**. We train a variant of RoSE that replaces the grayscale input with an RGB input (*i.e*, Ours w/ RGB). The performance drops by $0.79°$ on the LUCES benchmark. The result indicates the effectiveness of the grayscale input, which eliminates redundant chromatic information to enable accurate shading sequence estimation.

**Validation on ring-light setup**. We train RoSE using a different light path where the elevation decreases from $60°$ to $30°$ while rotating $360°$ around the $z$-axis. We denote this variant as "Ours w/ spiral". As shown in Table 6, this more complex light path leads to a performance drop (MAE of $17.60°$). This result highlights that the proposed ring-light setup is an effective and efficient design for predicting the shading sequence.

## 5 DISCUSSION

**Conclusion**. We propose RoSE, a novel method for monocular normal estimation that addresses the limitations of previous methods in their training paradigms to reduce 3D misalignment. By reformulating normal estimation to shading sequence estimation, RoSE facilitates normal estimation through an image-to-video generative model and a simple analytical solver. To further improve performance across more general scenarios, we train RoSE on MultiShade, a large-scale dataset with diverse materials and lighting conditions. Experiments show that RoSE outperforms state-of-the-art methods.

**Limitations & Future Work**. While RoSE demonstrates strong performance in normal estimation across various settings, it has several limitations. First, employing video diffusion models for shading sequence generation introduces additional computational overhead, which may limit the applicability of the method in real-time scenarios. Second, RoSE may struggle under extreme lighting conditions, particularly when large regions of the object receive insufficient illumination, resulting in degraded shading quality and less reliable normal predictions in those areas. Third, RoSE fails to produce high-quality normal maps on transparent or semi-transparent objects, and extending support for such cases will be an important direction for future work. Finally, the current evaluation is primarily object-centric, with a focus on robustness to varying light sources and reflectance properties. Extending RoSE to scene-centric settings remains an important direction for future work.

**Acknowledgment**. This work is supported by the Ministry of Education, Singapore, AcRF Tier 1 grant No. RG98/24, and ByteDance Inc.

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
