CONTENTS

## A   LLM Usage Statement

Large Language Models (LLMs) were only used in this research for writing optimization and grammar checking. No part of the theoretical contributions, experimental design, data analysis, or results was generated by LLMs.

## B   Broader Impact

The proposed method improves the accuracy of normal map estimation from a monocular image, with broad benefits across various applications. More precise geometry understanding can significantly improve downstream tasks such as 3D reconstruction, augmented reality, robotics, and digital content creation, enabling more immersive and interactive user experiences.

## C   Additional Implementation Details

**Training details**. As stated in the main paper, we build RoSE on top of SV3D (Voleti et al., 2024). During training, we initialize our model with pretrained SV3D weights (1.5B) and fine-tune the first convolutional layer as well as all parameters within the self-attention and cross-attention modules on the MultiShade dataset, which contains 90K objects (3.1M image–normal pairs), and evaluated on a validation set of 100 objects, each providing 100 image–normal pairs, result in 200M trainable parameters. Training is conducted for 80,000 steps with a learning rate of $1 \times 10^{-5}$ and a total batch size of 16. We use the AdamW optimizer with $\beta_1 = 0.9$, $\beta_2 = 0.999$, and $1 \times 10^{-8}$ for weight decay. Training is performed in float16 precision for efficiency, and we apply gradient clipping with a maximum norm of 1.0. The model is trained to predict a 9-frame shading sequence, with each frame at a resolution of $576 \times 576$. End-to-end training takes approximately one day on 8 NVIDIA H100 80GB GPUs.

**Testing details**. We follow the requirements specified in the baselines' inference code (He et al., 2024b; Fu et al., 2024; Ye et al., 2024; 2025; Bae & Davison, 2024) to prepare our test dataset, ensuring compatibility with each setup for a fair comparison. For both the LUCES (Mecca et al., 2021) and DiLiGenT (Shi et al., 2016) datasets, we use images indexed from 21 to 30 for testing, as the lights are more centered on the objects. All testing processes are performed on a single RTX A6000 Ada GPU. The total runtime includes the video diffusion model inference with 25 denoising steps and the shading to normal computation, the latter adding only a negligible cost of 0.045 seconds per object. For completeness, we also report the inference time of other methods for reference.

Table 7: Average inference time of monocular normal estimation methods per image (in seconds).

| Method | GeoWizard | DSINE | StableNormal | Lotus G | Lotus D | Neural LightRig | NiRNE | Ours |
|---|---|---|---|---|---|---|---|---|
| Time | 101.11 | 0.83 | 1.52 | 0.61 | 0.59 | 93.73 | 0.31 | 10.57 |

## D   Additional Details about MultiShade

**More Details about Material Augmentation**. We present a statistical comparison of the proposed dataset with other related datasets (He et al., 2024b; Ye et al., 2025; Jin et al., 2025; Ikehata, 2022; 2023) in Table 8, including works (He et al., 2024b; Ye et al., 2025) that are either recently released or not yet publicly available. We apply material augmentation (MA) with a probability of 0.5 by randomly replacing an object's material with one sampled from the MatSynth dataset (Vecchio & Deschaintre, 2024), selecting equally from metallic (617) or non-metallic (5,040) material groups. This process yields an additional 42,732 objects that share the same 3D geometry but differ in material appearance. The resulting MultiShade dataset, enriched with material diversity and rendered shading sequences, enables our method to achieve state-of-the-art performance on public benchmarks.

Table 8: Statistics of representative datasets used for normal estimation under arbitrary lighting. #$O$ and #$v$ denote the number of 3D models and rendered views, respectively. N.A. indicates that the corresponding information is not available. 'env.', 'par.', 'poi.' stand for environment light, parallel lights, and point lights, respectively.

| Dataset | #$O$ | #$v$ | Light Compose | Material |
|---|---|---|---|---|
| PS-Wild (Ikehata, 2023) | 410 | 1 | env.(31) /par. /poi. | AdobeStock (926) |
| PS-Mix (Ikehata, 2023) | 480 | 1 | env.(31) /par. /poi. | AdobeStock (897) |
| LightProp (He et al., 2024b) | 80K | 5 | env.(24) /area /poi. | Objaverse |
| RelitObjaverse (Jin et al., 2025) | 90K | 16 | env.(1,870) /area | Objaverse |
| DetailVerse (Ye et al., 2025) | 700K | 40 | N.A. | N.A. |
| **Ours** | **90K** | **6** | **env.(780) /par. /poi.** | **Objaverse + MatSynth (5,657)** |

**Rendering setup**. We construct our dataset using the Cycles rendering engine in Blender (Blender Online Community, 2018), selecting 90,546 filtered objects from Objaverse (Jin et al., 2025). Each object is rendered from six viewpoints. For each view, we implement one parallel light, one point light, or two HDR environment maps, selected from a pool of 780 real-world HDR environments (Poly Haven, 2025). The directions of the point and parallel lights are randomly sampled from the upper-front hemisphere facing the camera (see Fig. 7). The camera is positioned at a random distance $\tau$ between 1.5 and 1.8 meters from the object, with a focal length of 35 mm, following the setup in (Liu et al., 2023).

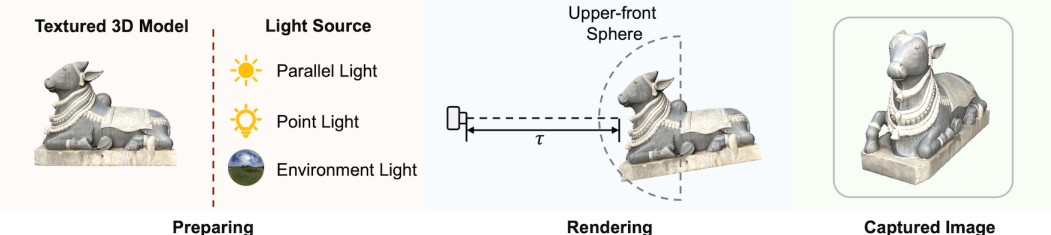

Figure 7: Image rendering setup.

# E    RESULTS ON WEB IMAGES

We present qualitative comparisons with state-of-the-art methods (NiRNE (Ye et al., 2025) and Neural LightRig (He et al., 2024b)) on additional images sourced from public resources, including the project page of StableNormal (Ye et al., 2024) and Google Images, as shown in Fig. 8 and Fig. 9. The surface reconstruction from normals is performed using the method from (Cao et al., 2022).

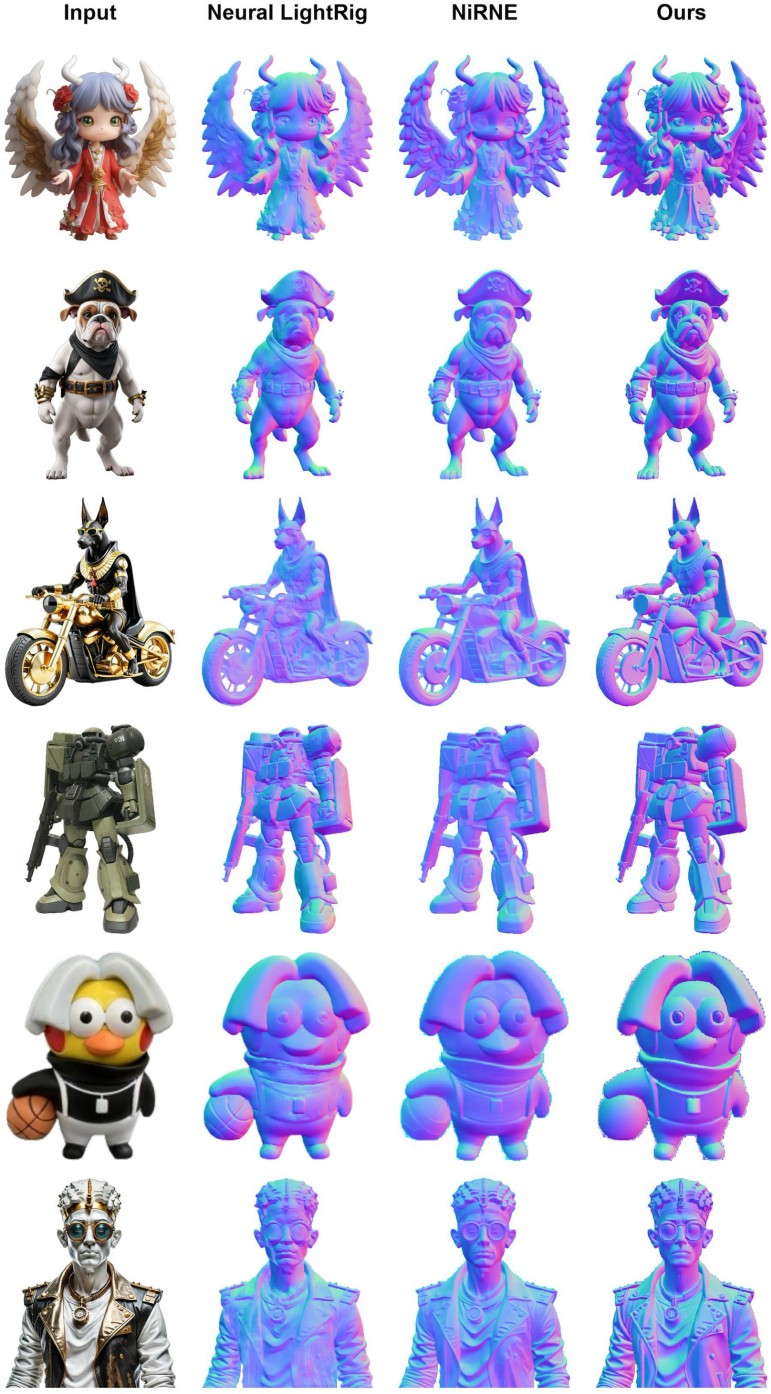

Figure 8: Qualitative comparison of normal maps on web images.

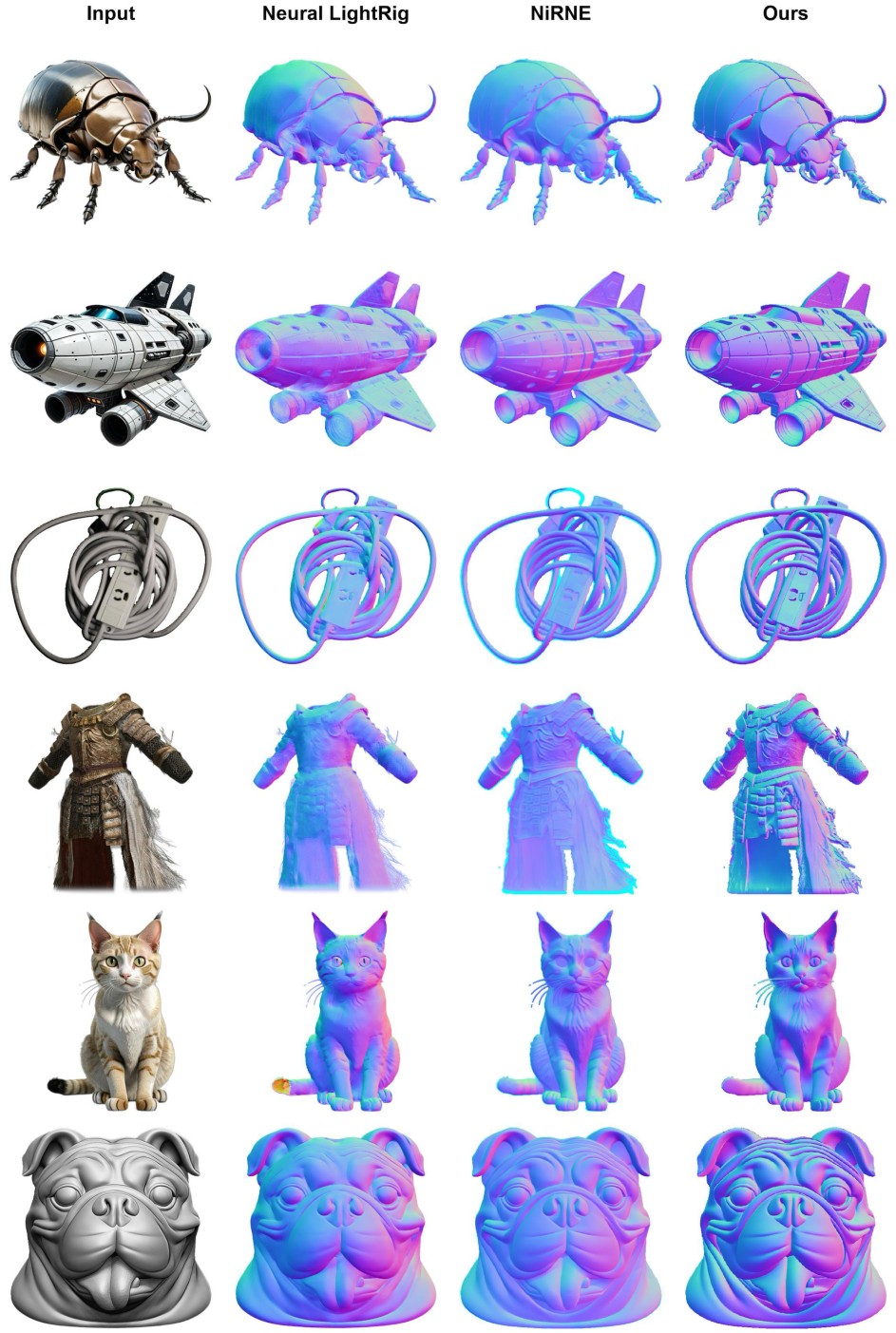

Figure 9: Qualitative comparison of normal maps on web images.

# F ADDITIONAL EXPERIMENT RESULTS ON POPULAR DATASETS

## F.1 RESULTS ON DILIGENT

We present a qualitative comparison of different methods for normal estimation. To avoid excessive redundancy, we select the normal map whose MAE is closest to the average MAE as a representative example for reference.

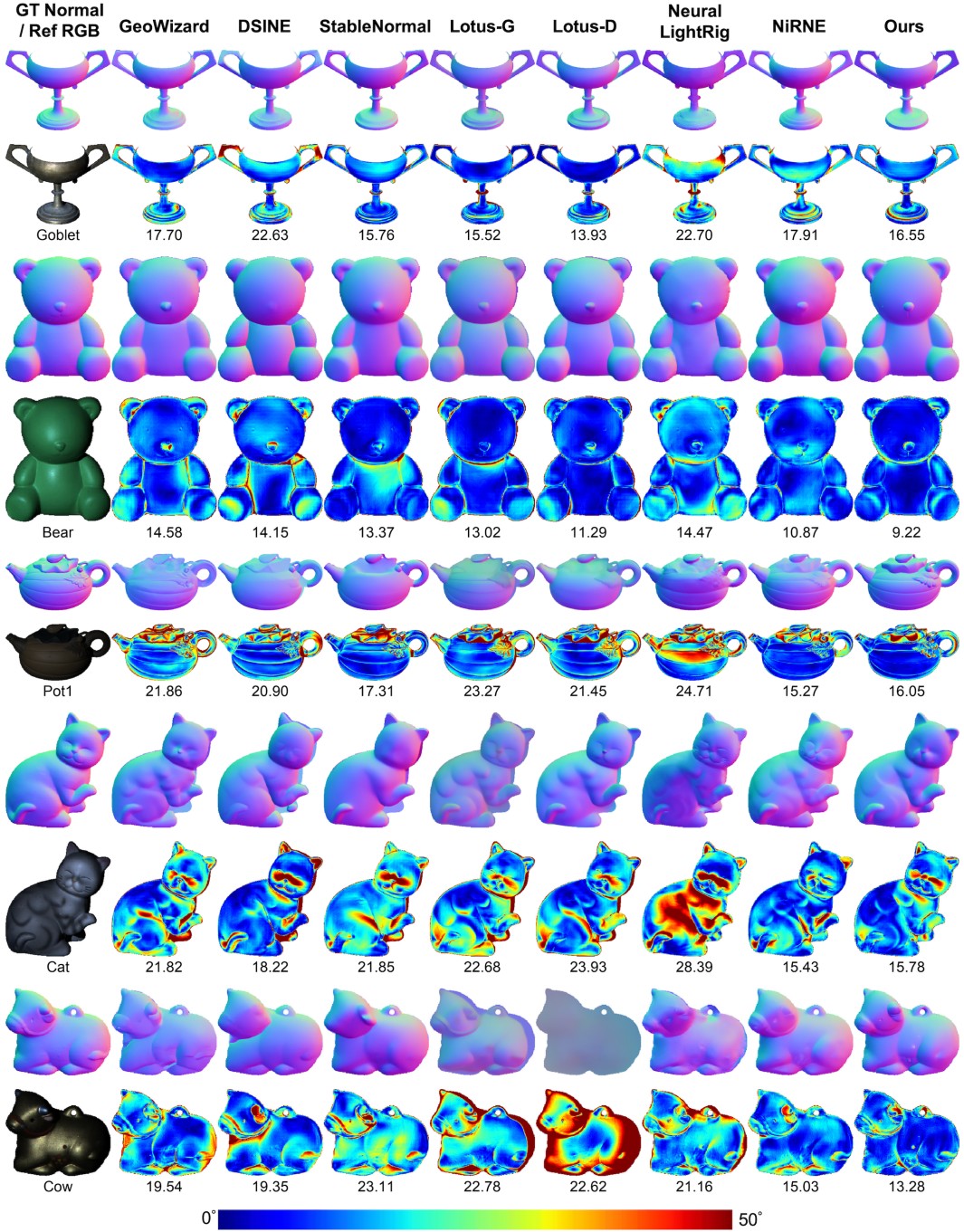

Figure 10: Qualitative comparison on normal maps and error maps for the GOBLET, BEAR, POT1, CAT, COW from the DiLiGenT (Shi et al., 2016) benchmark.

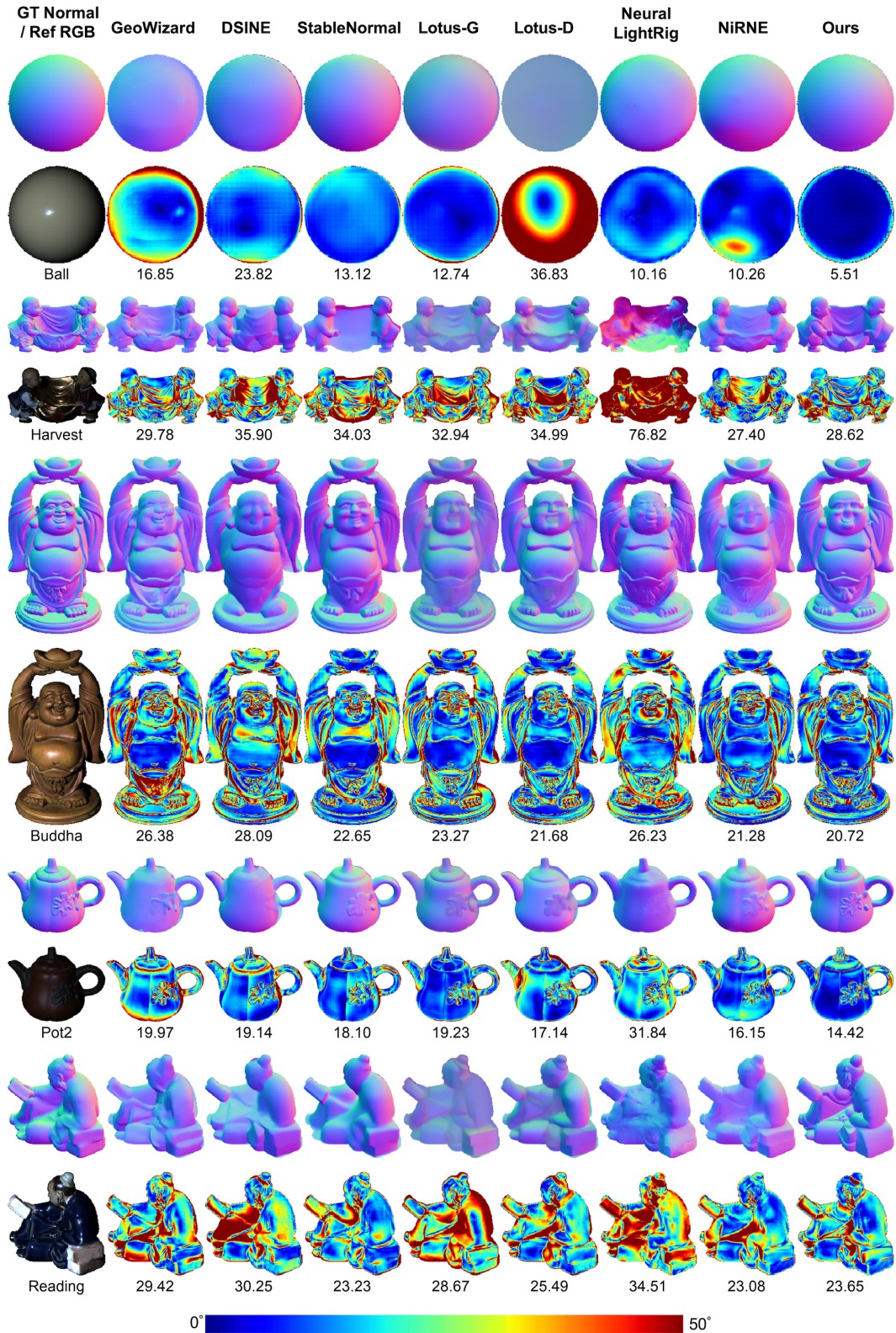

Figure 11: Qualitative comparison on normal maps and error maps for the BALL, HARVEST, BUDDHA, POT2, READING from the DiLiGenT (Shi et al., 2016) benchmark.

## F.2 RESULTS ON LUCES

We present a qualitative comparison on LUCES (Mecca et al., 2021) of different methods for normal estimation. To avoid excessive redundancy, we select the normal map whose MAE is closest to the average MAE as a representative example for reference.

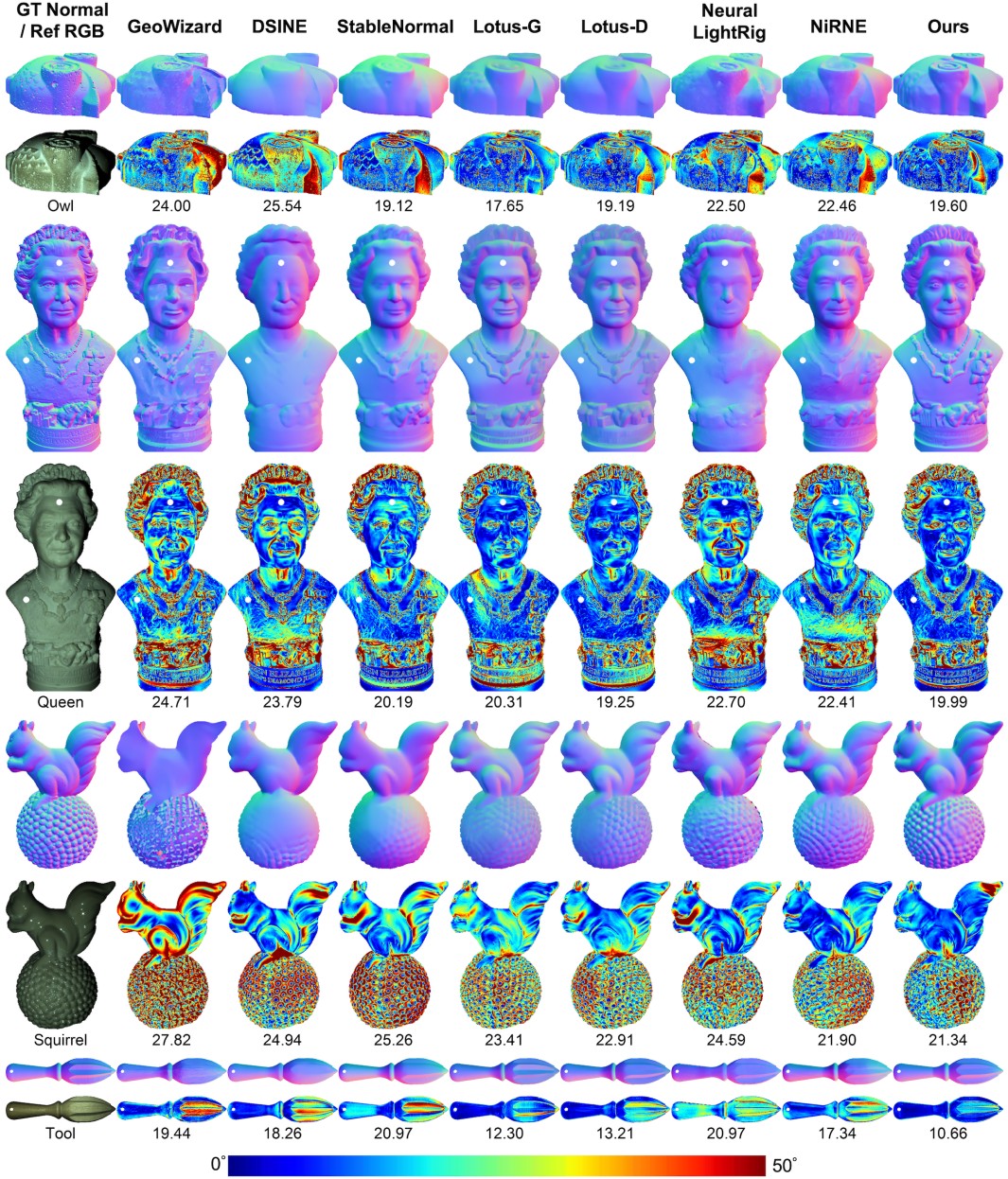

Figure 12: Qualitative comparison on normal maps and error maps for the OWL, QUEEN, SQUIRREL, TOOL from the LUCES (Mecca et al., 2021) benchmark.

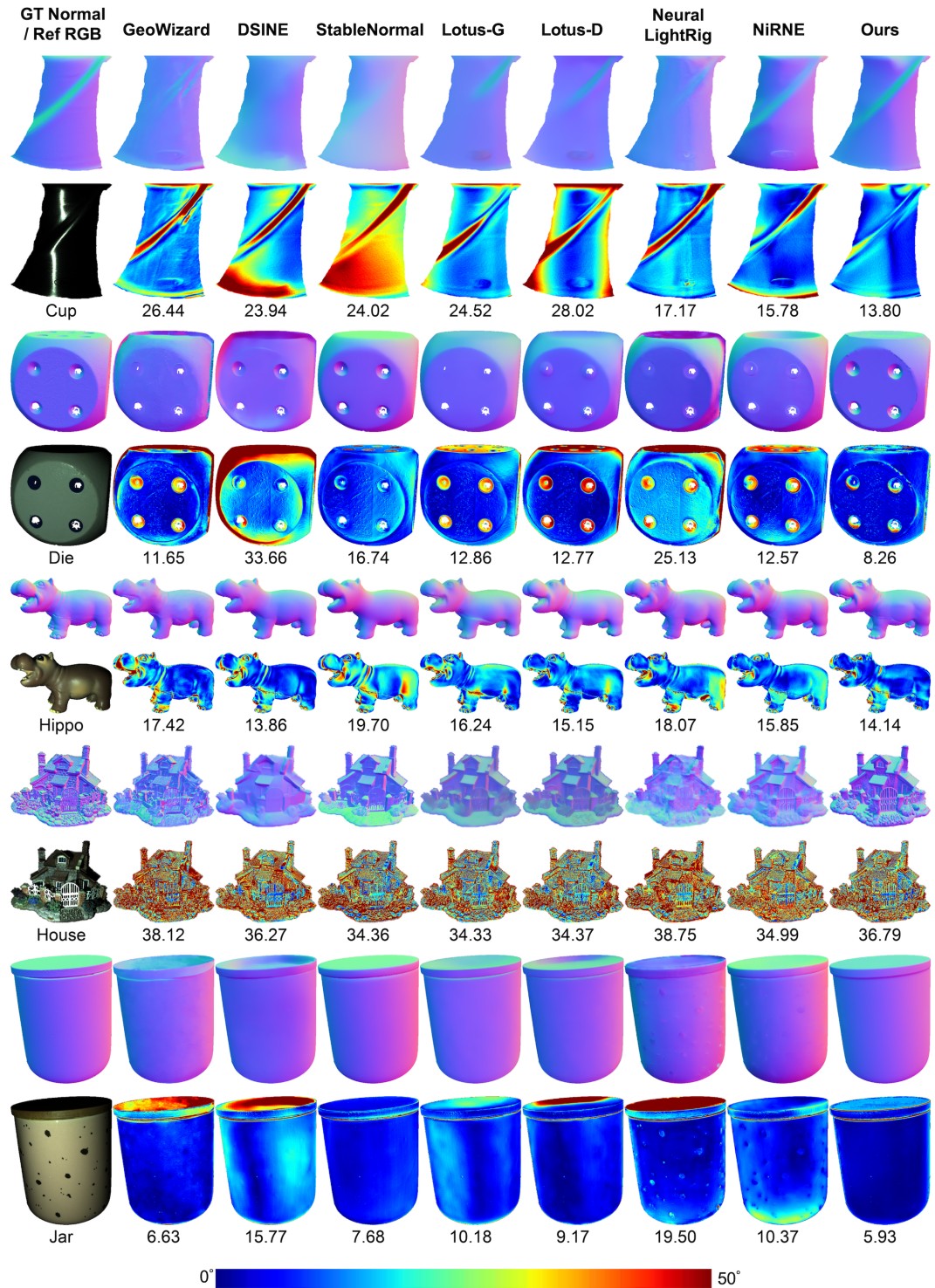

Figure 13: Qualitative comparison on normal maps and error maps for the CUP, DIE, HIPPO, HOUSE, and JAR from the LUCES (Mecca et al., 2021) benchmark.

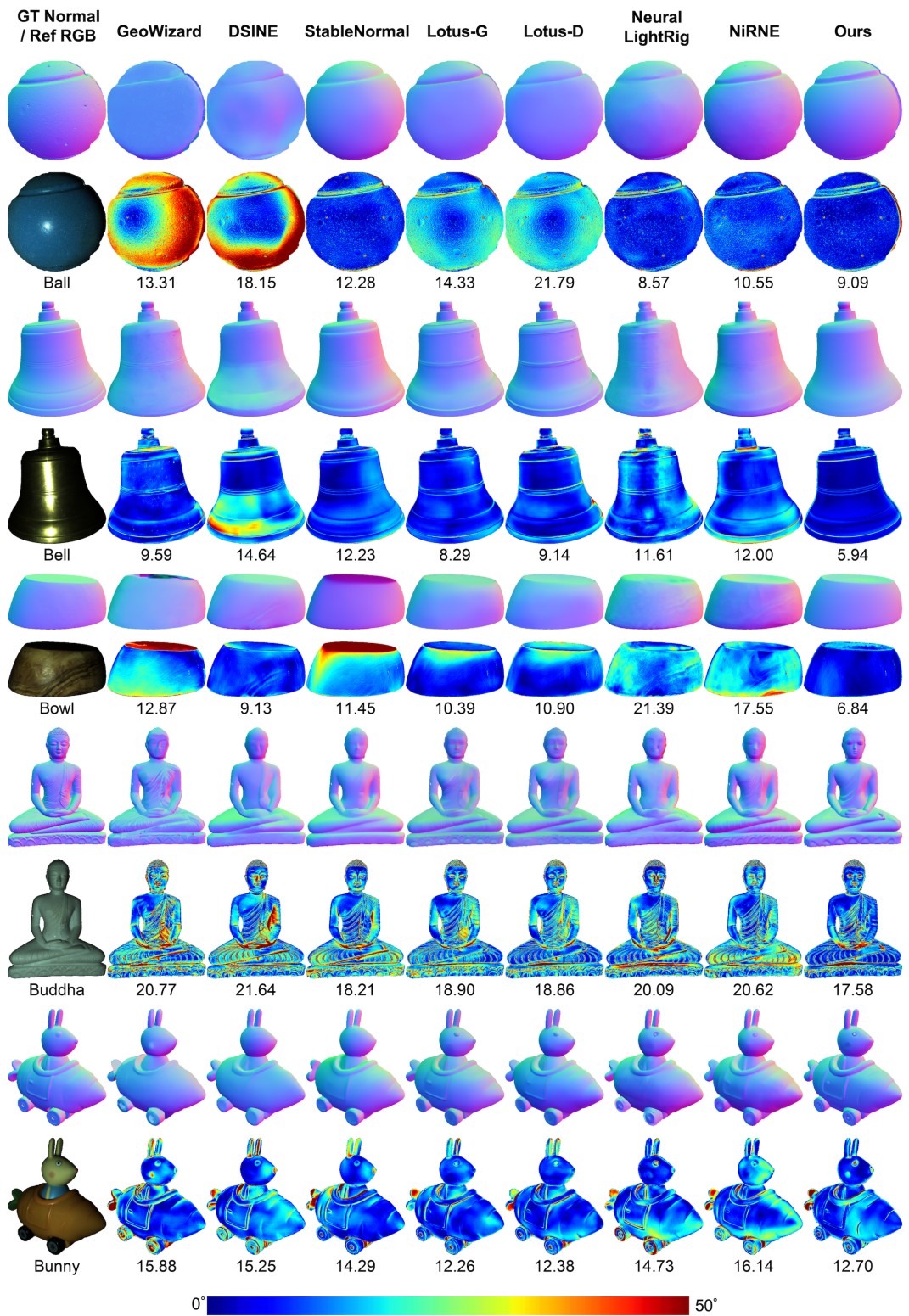

Figure 14: Qualitative comparison on normal maps and error maps for the OWL, QUEEN, SQUIRREL, TOOL from the LUCES (Mecca et al., 2021) benchmark.

**Qualitative analysis on shading sequence.**

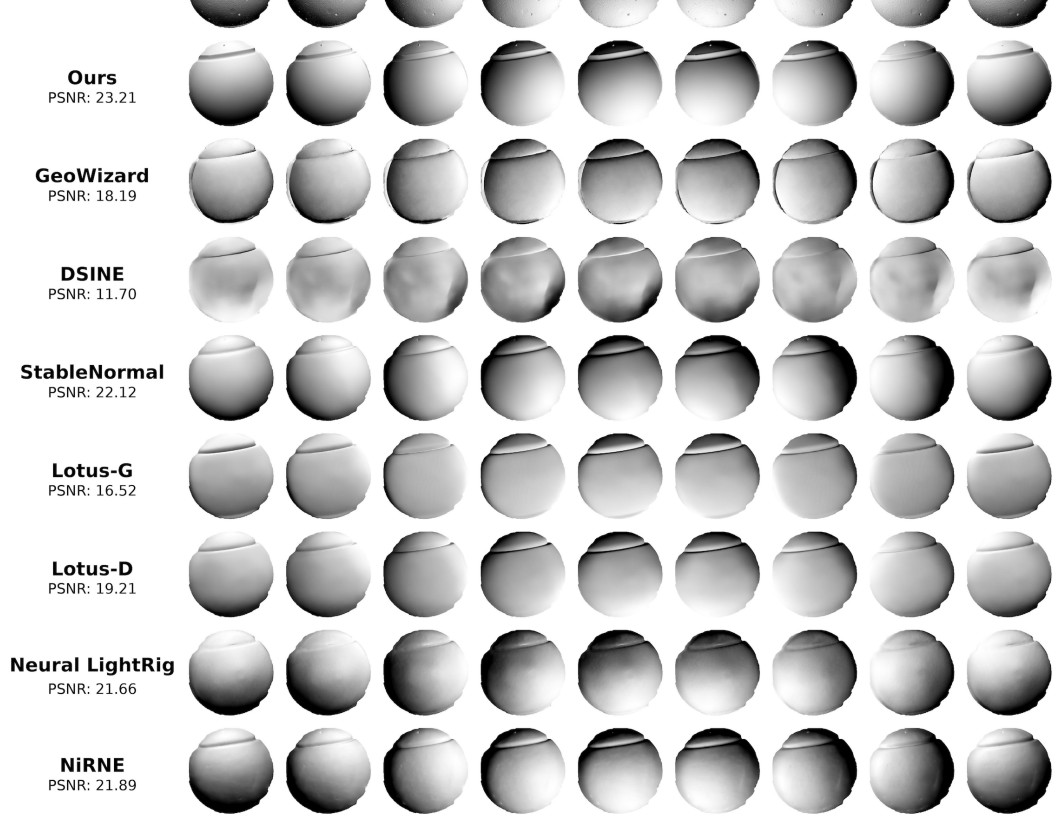

Figure 15: Qualitative comparison on shading sequence prediction for the BALL from LUCES (Mecca et al., 2021) benchmark.

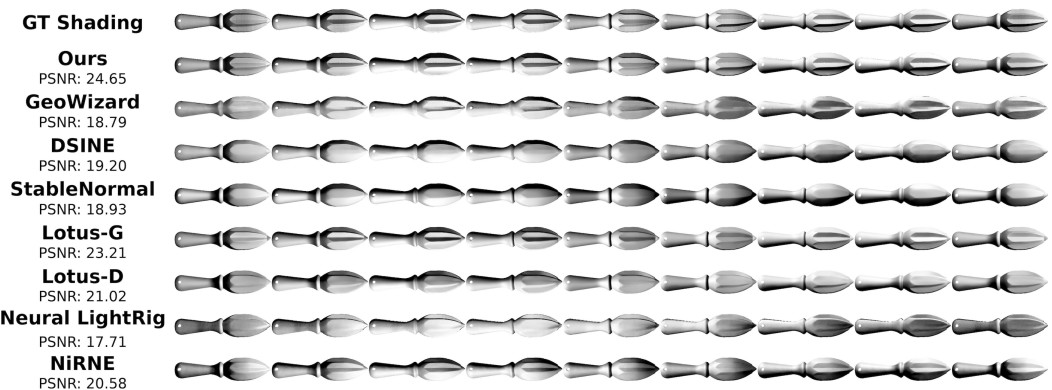

Figure 16: Qualitative comparison on shading sequence prediction for the TOOL from LUCES (Mecca et al., 2021) benchmark

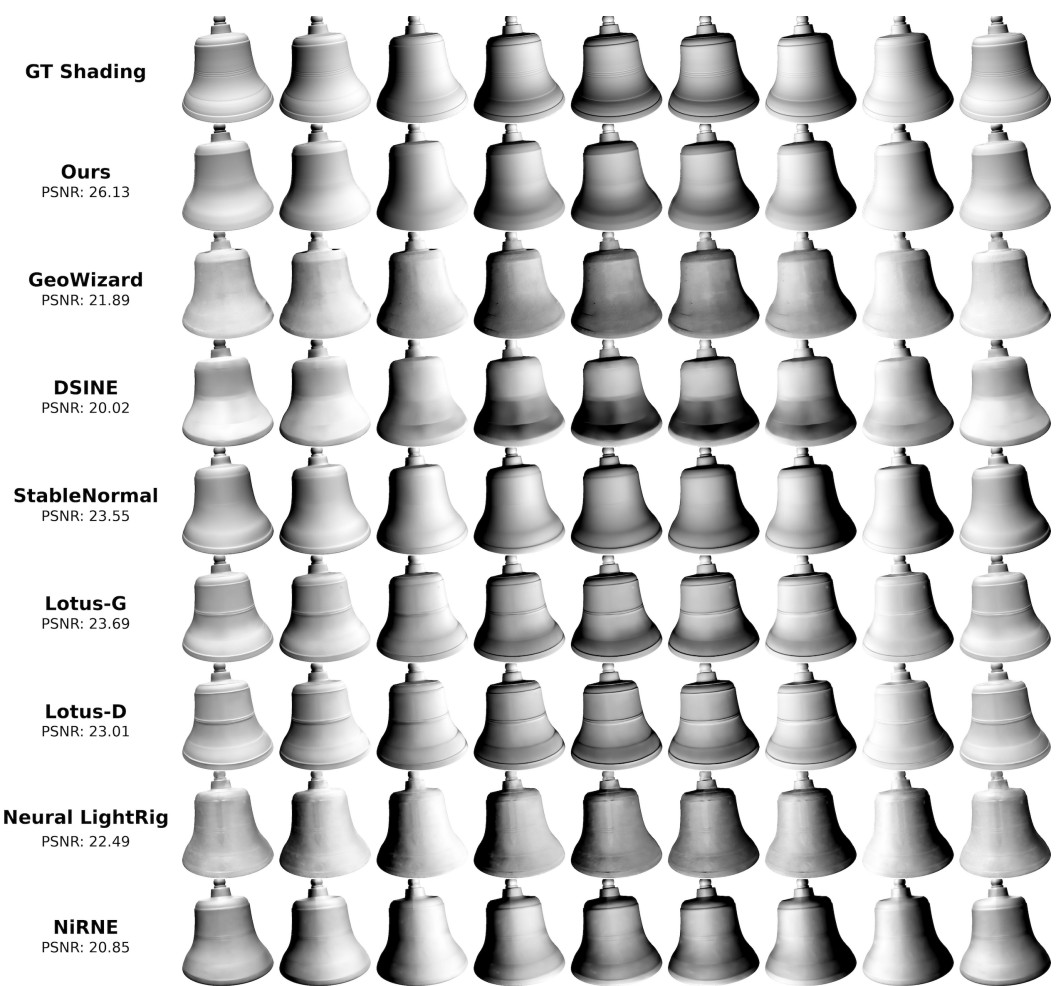

Figure 17: Qualitative comparison on shading sequence prediction for the BELL from LUCES (Mecca et al., 2021) benchmark

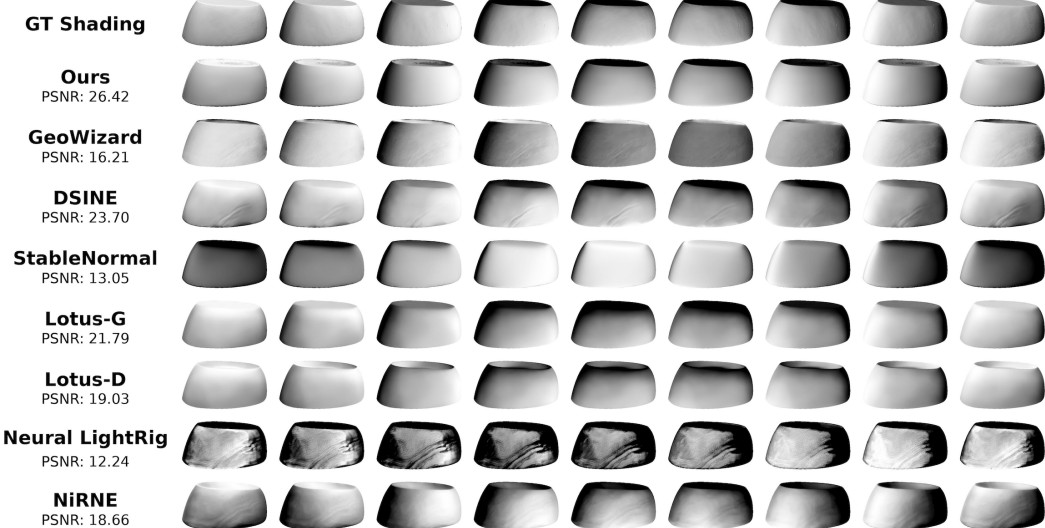

Figure 18: Qualitative comparison on shading sequence prediction for the BOWL from LUCES (Mecca et al., 2021) benchmark

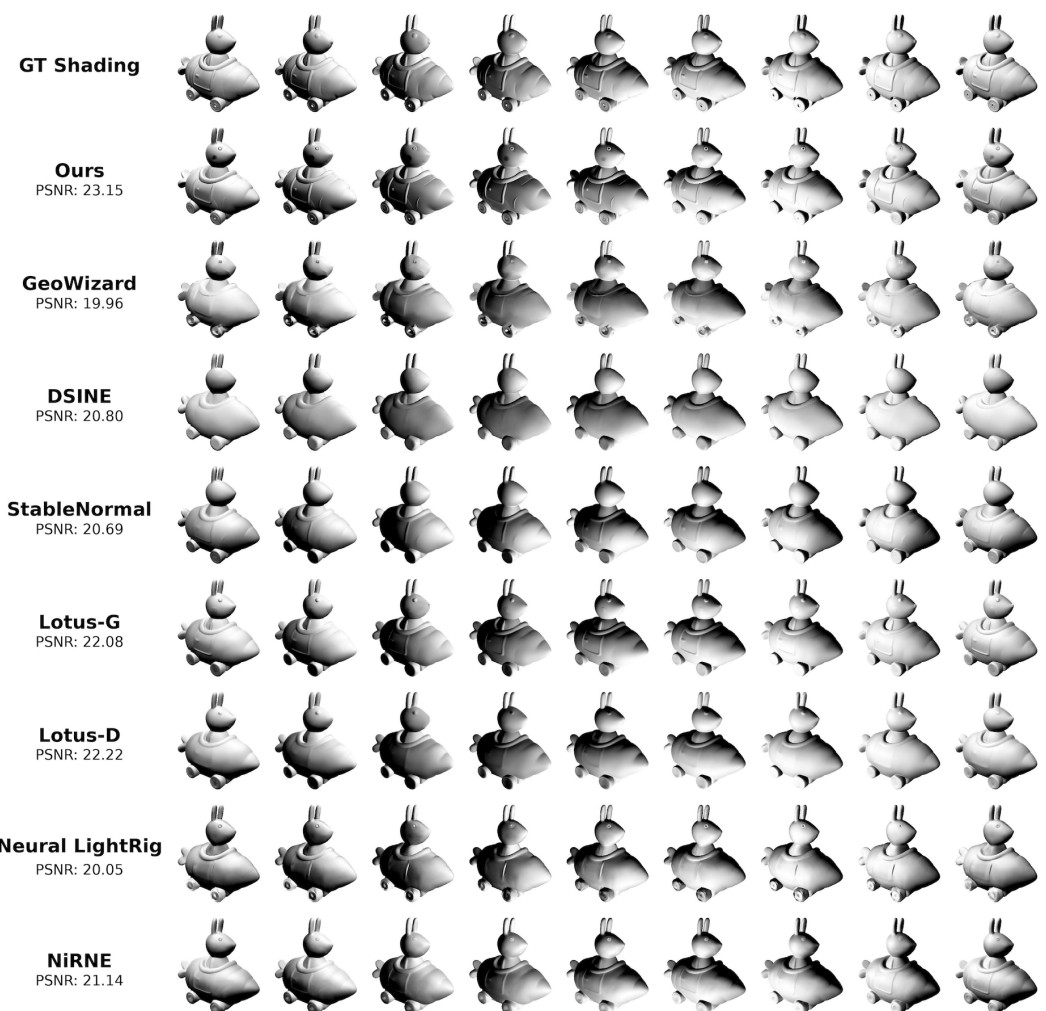

Figure 19: Qualitative comparison on shading sequence prediction for the BUNNY from LUCES (Mecca et al., 2021) benchmark

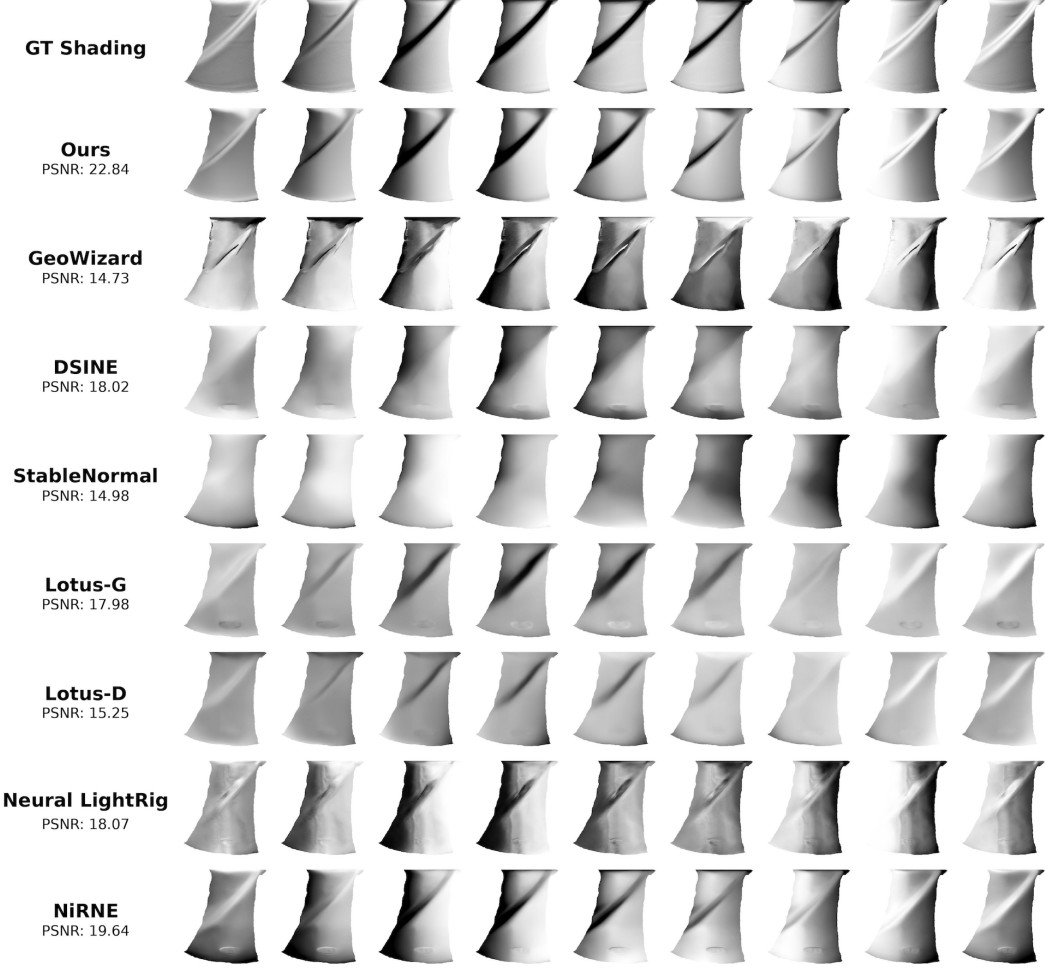

Figure 20: Qualitative comparison on shading sequence prediction for the CUP from LUCES (Mecca et al., 2021) benchmark

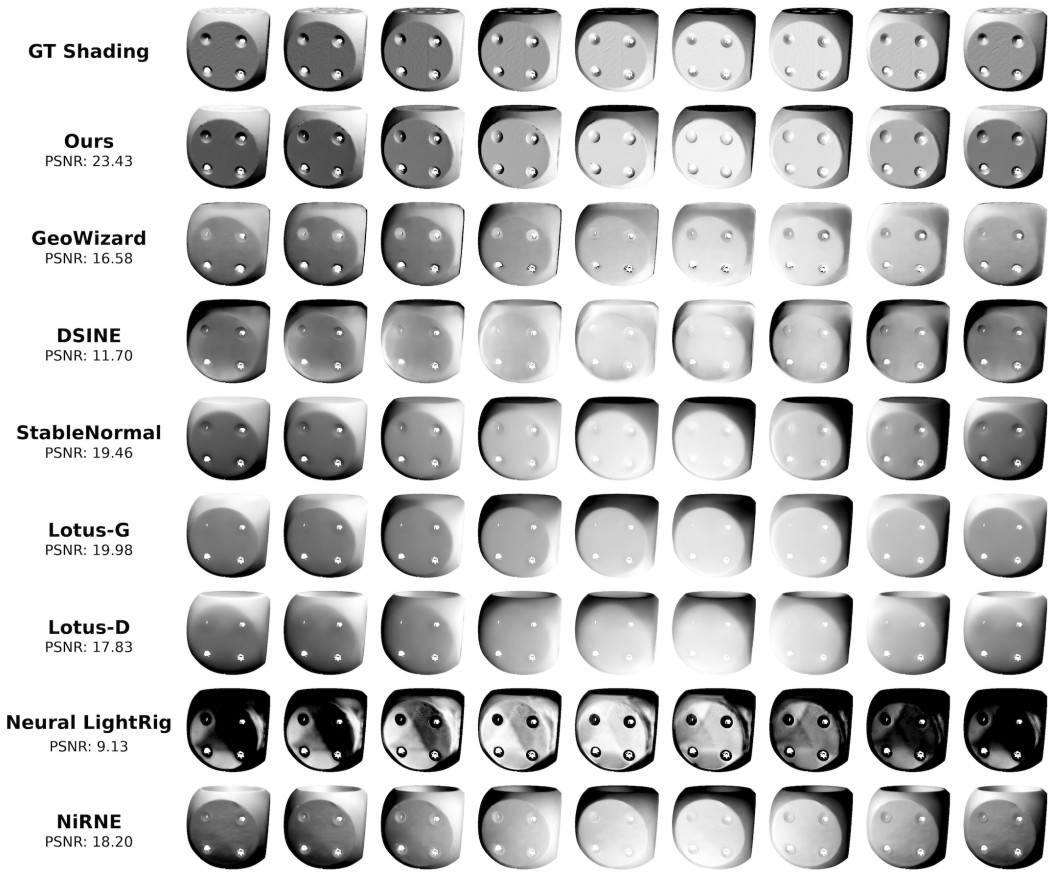

Figure 21: Qualitative comparison on shading sequence prediction for the DIE from LUCES (Mecca et al., 2021) benchmark

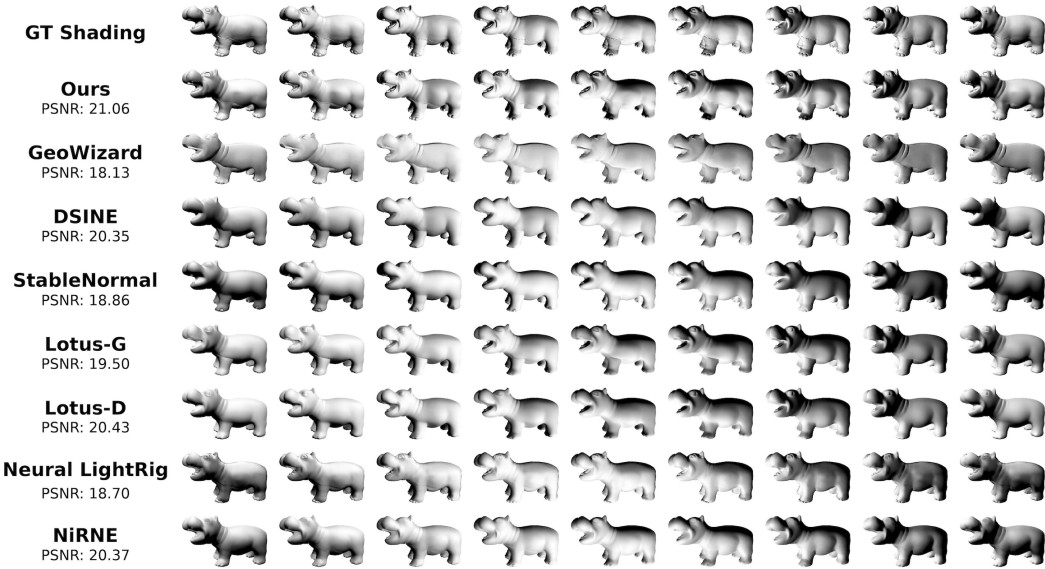

Figure 22: Qualitative comparison on shading sequence prediction for the HIPPO from LUCES (Mecca et al., 2021) benchmark

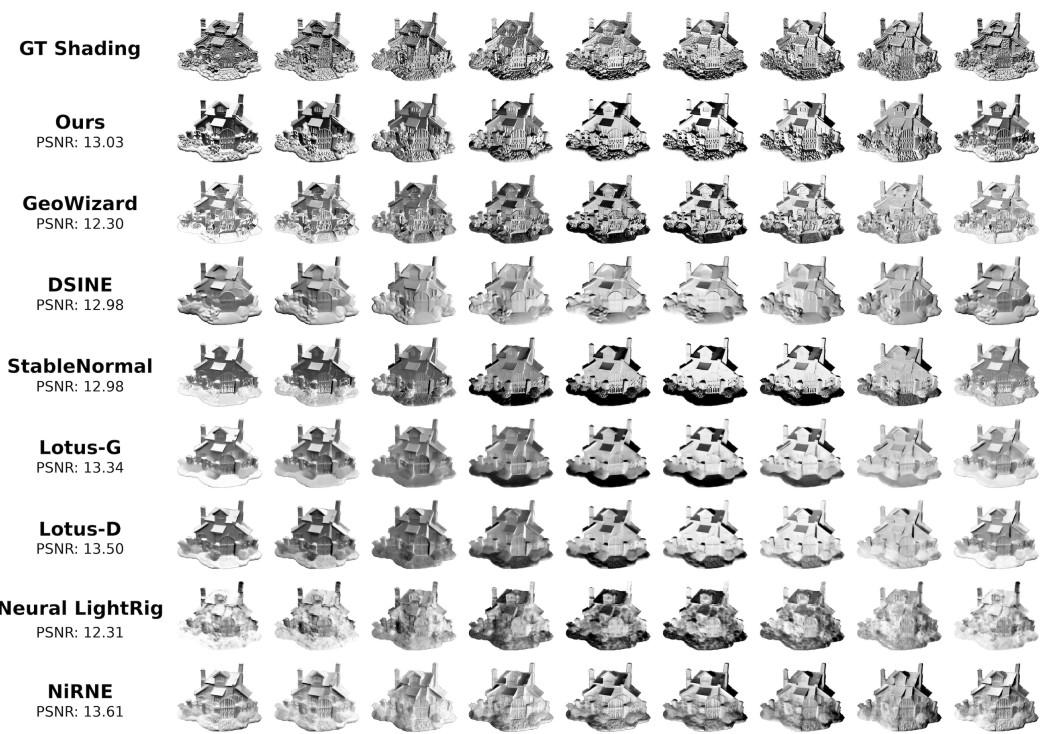

Figure 23: Qualitative comparison on shading sequence prediction for the HOUSE from LUCES (Mecca et al., 2021) benchmark

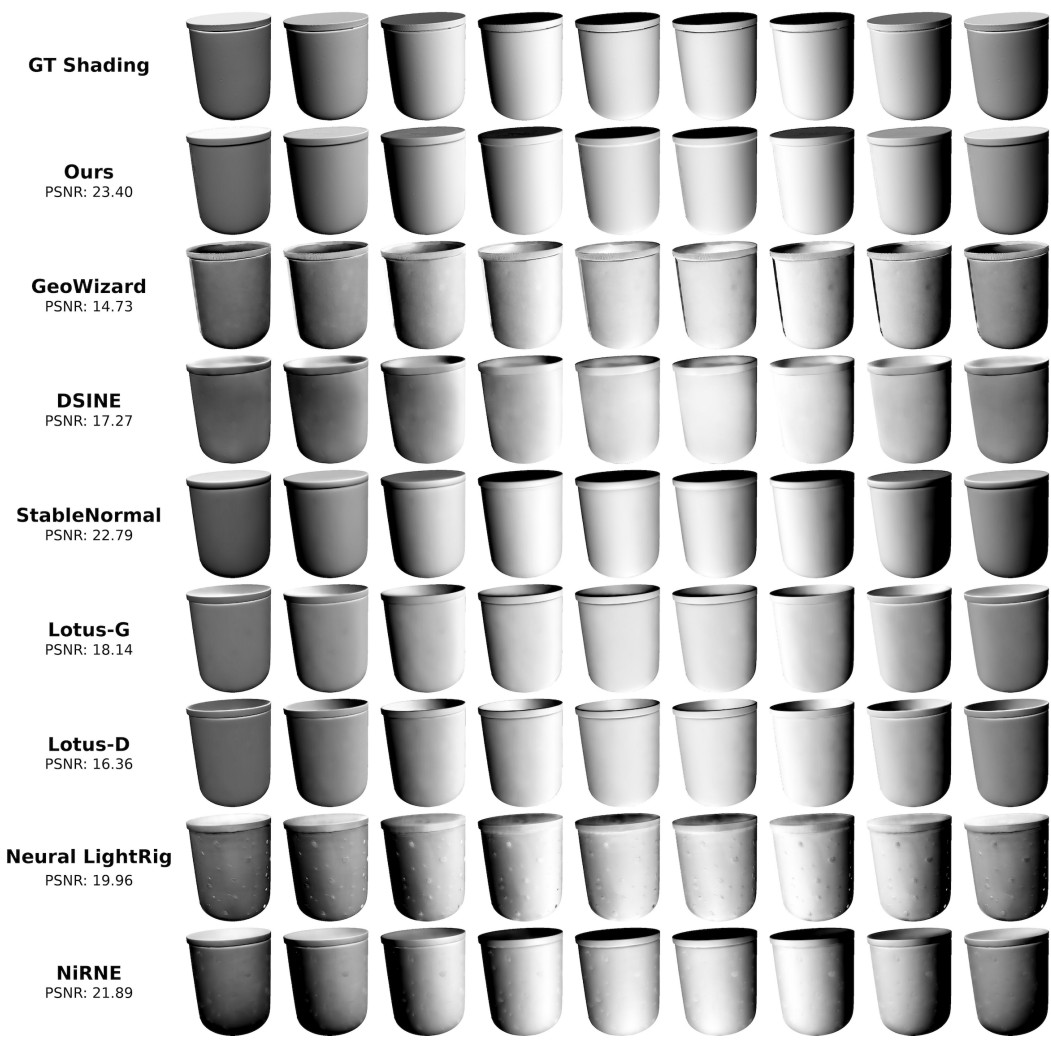

Figure 24: Qualitative comparison on shading sequence prediction for the JAR from LUCES (Mecca et al., 2021) benchmark

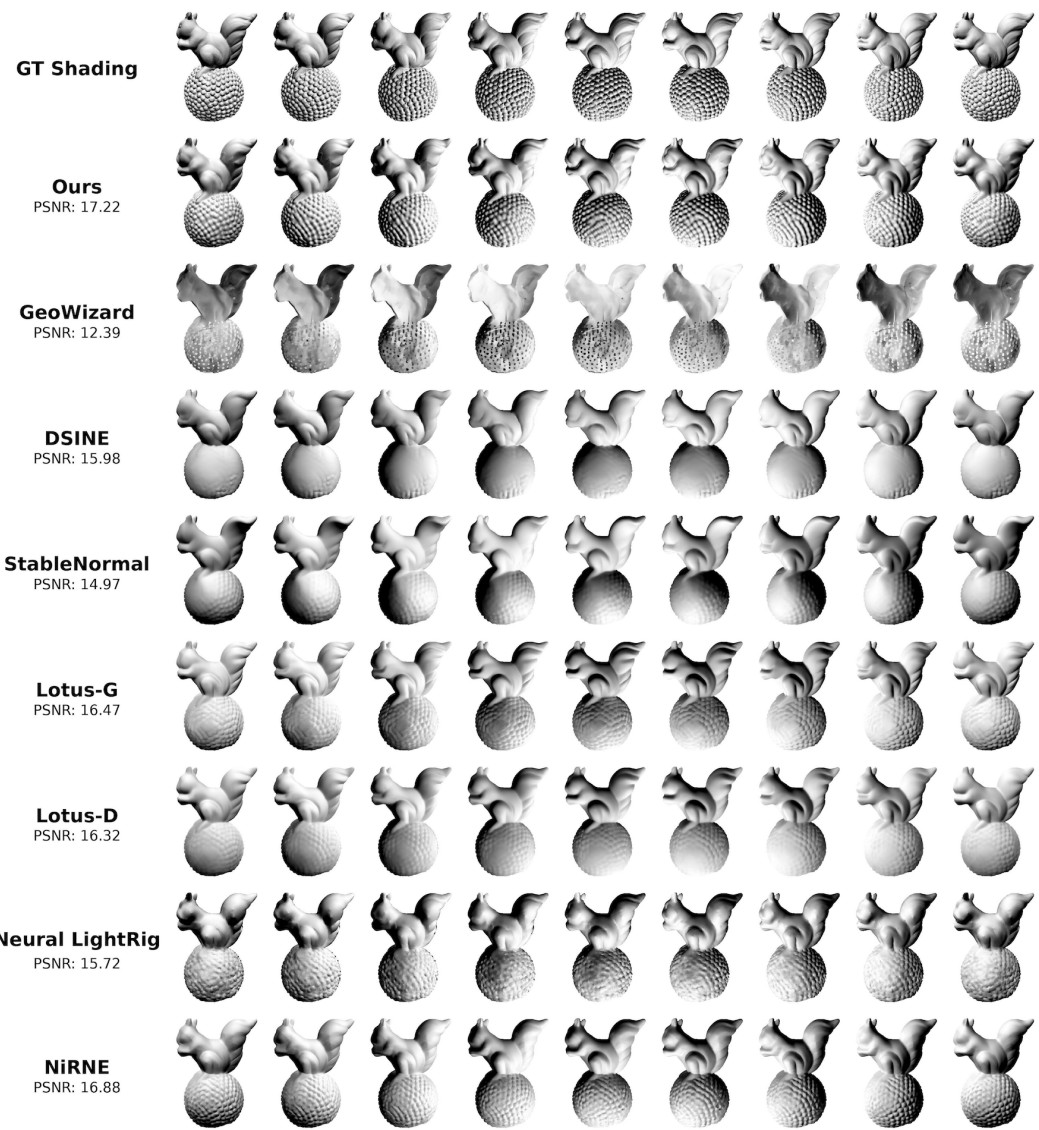

Figure 25: Qualitative comparison on shading sequence prediction for the SQUIRREL from LUCES (Mecca et al., 2021) benchmark

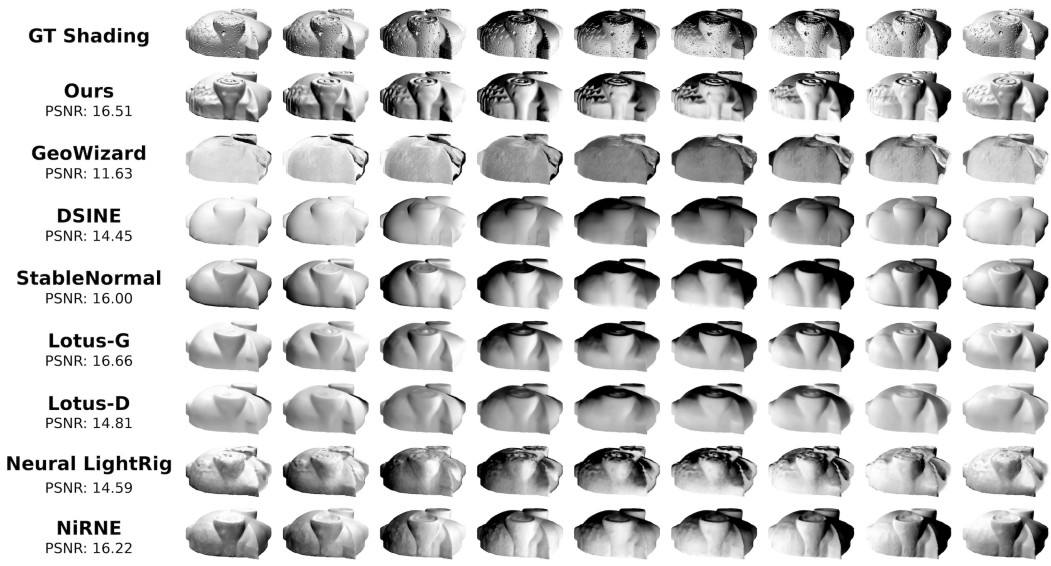

Figure 26: Qualitative comparison on shading sequence prediction for the OWL from LUCES (Mecca et al., 2021) benchmark

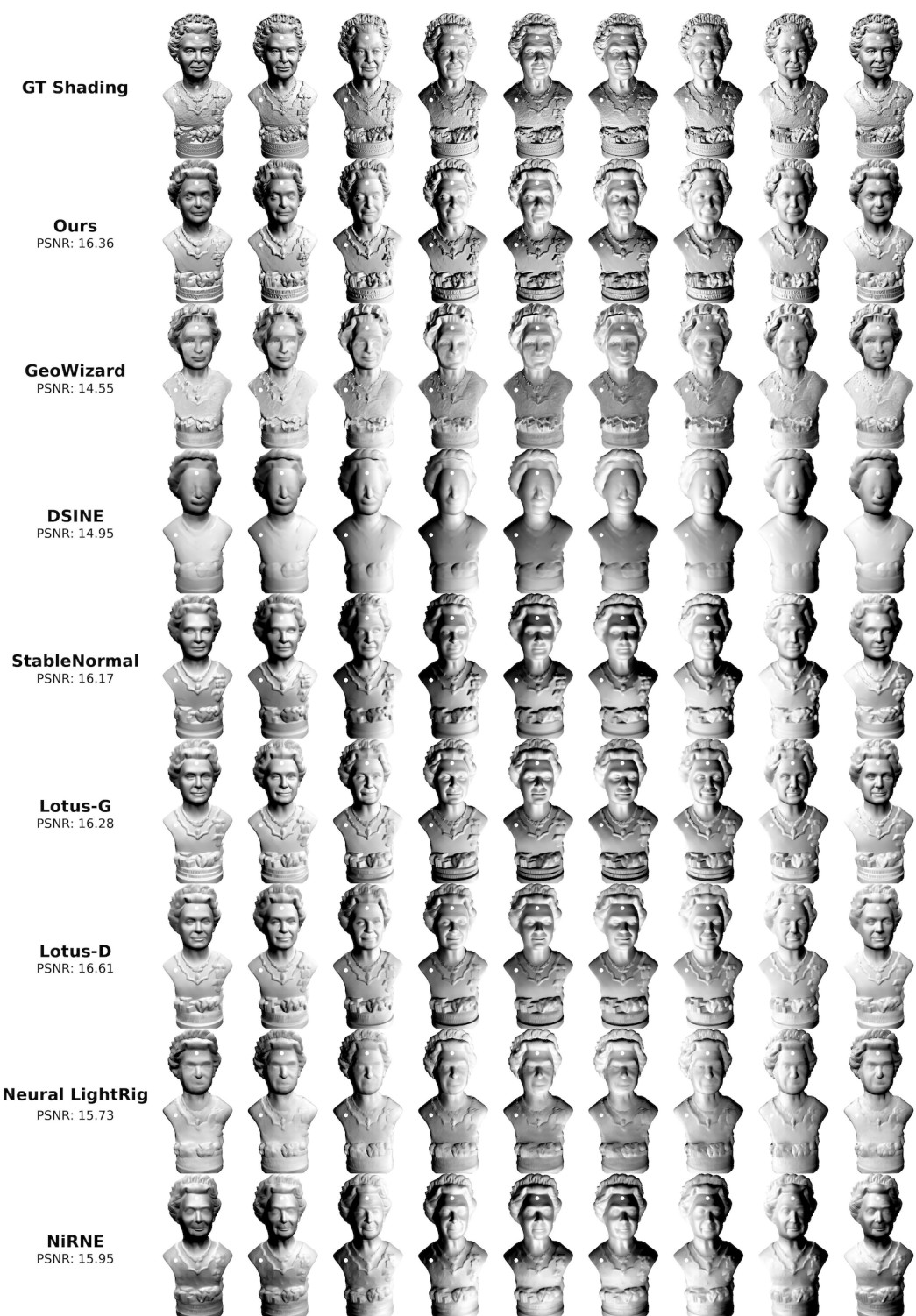

Figure 27: Qualitative comparison on shading sequence prediction for the QUEEN from LUCES (Mecca et al., 2021) benchmark

## F.3 Results on LightProp

We report quantitative results on the LightProp dataset (He et al., 2024b). Our method achieves the second-best overall performance in terms of mean and median angular errors for normal map estimation. We re-ran the entire evaluation, as we observed discrepancies between our results and those reported in the original paper (He et al., 2024b). The metric computation scripts used in our evaluation follow the implementation in (Bae & Davison, 2024)[2].

Table 9: Quantitative comparison in terms of **Mean** and **Median** Angular Errors of the normal map on LightProp test set, and the percentage of objects below a specific error bound. **Bold** (underline) numbers indicate the **best** (second-best) results among single-view normal estimation methods.

| Method | Mean ↓ | Median ↓ | 3°(%) ↑ | 5°(%) ↑ | 7.5°(%) ↑ | 11.25°(%) ↑ | 22.5°(%) ↑ | 30°(%) ↑ |
|---|---|---|---|---|---|---|---|---|
| GeoWizard | 21.03 | 13.07 | 9.94 | 20.29 | 31.63 | 44.87 | 68.23 | 76.97 |
| DSINE | 22.16 | 14.02 | 9.48 | 18.00 | 28.31 | 41.85 | 66.99 | 76.12 |
| StableNormal | 19.66 | 12.98 | 3.78 | 10.64 | 23.15 | 42.50 | 73.85 | 82.37 |
| Lotus-D | 19.10 | 12.26 | 10.26 | 20.97 | 32.51 | 46.83 | 71.83 | 80.45 |
| Lotus-G | 19.19 | 12.15 | 10.91 | 22.06 | 33.91 | 47.35 | 71.12 | 79.70 |
| Neural LightRig | **15.29** | **8.84** | **20.13** | **32.33** | **44.68** | **57.99** | **78.99** | **85.79** |
| NiRNE | 17.87 | 12.38 | 7.21 | 16.69 | 29.13 | 45.68 | 75.01 | 84.02 |
| Ours | 17.40 | 11.00 | 17.15 | 26.49 | 37.33 | 50.79 | 75.29 | 83.10 |

---

[2]https://github.com/baegwangbin/DSINE/blob/main/projects/dsine/test.py

## F.4 RESULTS ON NATURAL LIGHT PHOTOMETRIC STEREO DATASET

We conduct quantitative comparison results on a synthetic dataset (NaPS) proposed in (Li et al., 2024b), which contains a series of rendered images based on objects selected from (Shi et al., 2016), under varying environment lighting, material properties, and object shapes. The dataset is organized into four groups: (1) the light group, which evaluates performance on a single object under different lighting conditions; (2) the shape group, which compares different object geometries; (3) the reflectance group, which assesses performance on diffuse and specular materials; and (4) the spatially varying material group, which poses a challenging scenario with complex, spatially varying materials. During testing, we select the 10 images with the highest average brightness for evaluation. Our method achieves either the best or second-best performance in each group and ranks first in terms of overall average performance ($11.35°$, ours vs. $12.32°$, NiRNE, the second best method).

Table 10: Results on the NaPS (Li et al., 2024b) dataset for normal estimation under natural lighting. 'L., A., S., U.' denote four types of environment maps, Landscape, Attic, Studio, and Urban, covering both outdoor and indoor settings. 'D., S.' in the reflectance and spatially varying material groups refer to diffuse and specular materials, respectively. Numbers indicate the MAE ($\downarrow$) of the estimated normal maps. **Bold** (underline) numbers indicate the **best** (second-best) results.

| Method | Light Group | | | | | Reflectance Group | | | | |
|---|---|---|---|---|---|---|---|---|---|---|
| | Cow (L.) | Cow (A.) | Cow (S.) | Cow (U.) | AVG | Pot2 (D.) | Pot2 (S.) | Reading (D.) | Reading (S.) | AVG |
| GeoWizard | 10.24 | 10.59 | 12.24 | 11.52 | 11.15 | 12.11 | 11.43 | 16.09 | 14.18 | 13.45 |
| DSINE | 14.41 | 15.65 | 14.41 | 13.43 | 14.48 | 15.86 | 14.63 | 17.03 | 17.23 | 16.19 |
| StableNormal | 14.58 | 12.49 | 16.11 | 16.61 | 14.95 | 10.44 | 11.24 | 14.87 | 13.54 | 12.52 |
| Lotus-D | **8.43** | **9.34** | 10.82 | **10.00** | **9.65** | 10.00 | 10.03 | 14.53 | 13.14 | 11.92 |
| Lotus-G | 11.66 | 11.28 | 12.48 | 10.29 | 11.43 | 12.00 | 11.51 | 15.93 | 13.67 | 13.28 |
| Neural LightRig | 9.25 | 9.90 | 11.98 | 11.21 | 10.59 | 10.54 | 10.22 | 13.73 | 12.63 | 11.78 |
| NiRNE | 9.89 | 10.66 | 12.72 | 10.55 | 10.95 | **7.66** | **8.66** | **11.21** | **10.81** | **9.58** |
| Ours | 9.85 | 10.06 | **10.51** | 10.09 | 10.13 | 10.74 | 9.77 | 11.73 | 11.07 | 10.83 |
| Method | Shape Group | | | | | Spatially Varying Material Group | | | | |
| | Ball | Bear | Buddha | Reading | AVG | Pot2 (D.) | Pot2 (S.) | Reading (D.) | Reading (S.) | AVG |
| GeoWizard | 3.93 | 9.71 | 21.52 | 15.36 | 12.63 | 14.42 | 14.15 | 22.82 | 22.32 | 18.43 |
| DSINE | 27.68 | 9.85 | 24.22 | 16.45 | 19.55 | 23.92 | 20.19 | 22.81 | 21.22 | 22.04 |
| StableNormal | 8.33 | **8.04** | 16.77 | 13.64 | 11.69 | 15.71 | 15.10 | 19.77 | 18.90 | 17.37 |
| Lotus-D | 8.71 | 9.30 | **16.15** | 13.50 | 11.91 | 13.18 | 13.54 | 21.95 | 17.86 | 16.63 |
| Lotus-G | 11.18 | 9.51 | 16.64 | 14.46 | 12.95 | 16.20 | 15.27 | 24.59 | 18.15 | 18.55 |
| Neural LightRig | **3.11** | 9.18 | 17.37 | 13.88 | 10.89 | 13.72 | 14.30 | 23.85 | 24.23 | 19.03 |
| NiRNE | 8.57 | 9.45 | 18.37 | 11.92 | 12.08 | 14.04 | 14.42 | 19.11 | 19.10 | 16.67 |
| Ours | 5.25 | 8.71 | 17.56 | **11.23** | **10.69** | **11.98** | **11.87** | **16.06** | **15.05** | **13.74** |

### F.5 OVERALL PERFORMANCE COMPARISON

In this paper, we conduct a comprehensive analysis across multiple benchmark datasets, including two real-world datasets (DiLiGenT (Shi et al., 2016) and LUCES (Mecca et al., 2021)) and three synthetic datasets (MultiShade, NaPS (Li et al., 2024b), and LightProp (He et al., 2024b)). According to Table 11, our method achieves the best average performance in terms of MAE and overall ranking. This demonstrates the strong generalization ability of the proposed RoSE across diverse scenarios, lighting conditions, and object types.

Table 11: Quantitative comparison over all five datasets. We report the MAE (↓) over all objects and the rank (↓) among methods at specific dataset for comparison.

| Method | DiLiGenT | | LUCES | | MultiShade | | LightProp | | NaPS | | AVG | |
|---|---|---|---|---|---|---|---|---|---|---|---|---|
| | MAE(↓) | Rank(↓) | MAE(↓) | Rank(↓) | MAE(↓) | Rank(↓) | MAE(↓) | Rank(↓) | MAE(↓) | Rank(↓) | MAE(↓) | Rank(↓) |
| GeoWizard | 21.79 | 5.00 | 22.49 | 8.00 | 20.46 | 6.00 | 21.03 | 7.00 | 13.91 | 5.00 | 19.94 | 6.20 |
| DSINE | 23.25 | 7.00 | 21.82 | 7.00 | 22.53 | 8.00 | 22.16 | 8.00 | 18.06 | 8.00 | 21.56 | 7.60 |
| StableNormal | 20.44 | 3.00 | 20.34 | 5.00 | 19.71 | 5.00 | 19.66 | 6.00 | 14.13 | 7.00 | 18.86 | 5.20 |
| Lotus-D | 22.94 | 6.00 | 18.56 | 4.00 | _18.48_ | _2.00_ | 19.10 | 4.00 | 12.53 | 3.00 | 18.32 | 3.80 |
| Lotus-G | 21.41 | 4.00 | _17.44_ | _2.00_ | 18.76 | 3.00 | 19.19 | 5.00 | 14.05 | 6.00 | 18.10 | 4.00 |
| Neural LightRig | 29.10 | 8.00 | 20.95 | 6.00 | 20.59 | 7.00 | **15.29** | **1.00** | 13.07 | 4.00 | 19.80 | 5.20 |
| NiRNE | _17.27_ | _2.00_ | 17.88 | 3.00 | 19.57 | 4.00 | 17.87 | 3.00 | _12.32_ | _2.00_ | _16.98_ | _2.80_ |
| Ours | **16.36** | **1.00** | **14.48** | **1.00** | **15.37** | **1.00** | _17.40_ | _2.00_ | **11.35** | **1.00** | **14.99** | **1.20** |

# G  CASE ANALYSIS

We conduct a detailed analysis across variations in lighting, textures, and BRDFs through three groups of controlled experiments. The first group examines how different lighting conditions, including simple light, parallel light, point light, and challenging light, influence performance, with texture and BRDF fixed. The second group evaluates the impact of texture complexity by comparing a simple texture with three spatially varying textures, while holding lighting and BRDF constant. The third group isolates material effects by varying the BRDF under simple lighting and texture settings. The results (see Fig. 28-Fig. 30) show that our method consistently produces high-quality normal maps relative to other approaches. However, performance does degrade under highly complex textures, challenging lighting conditions, or difficult material properties.

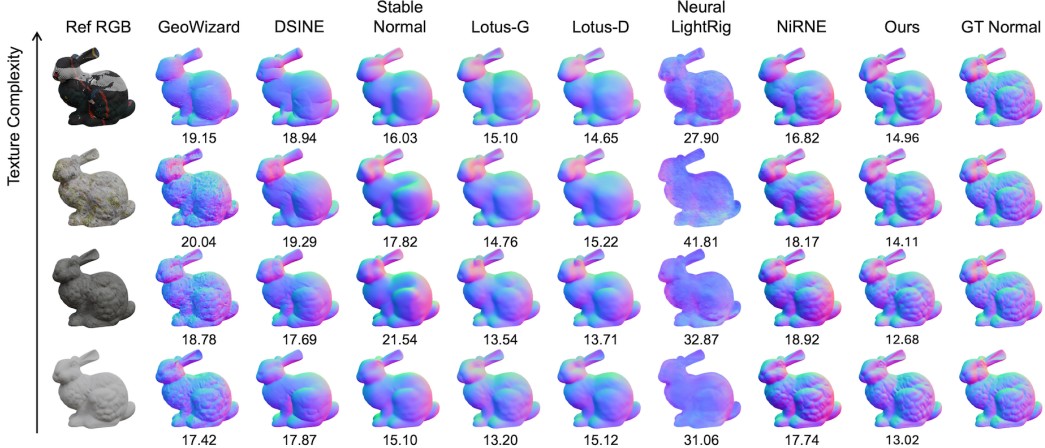

Figure 28: Analysis of each method's performance across different texture settings on BUNNY.

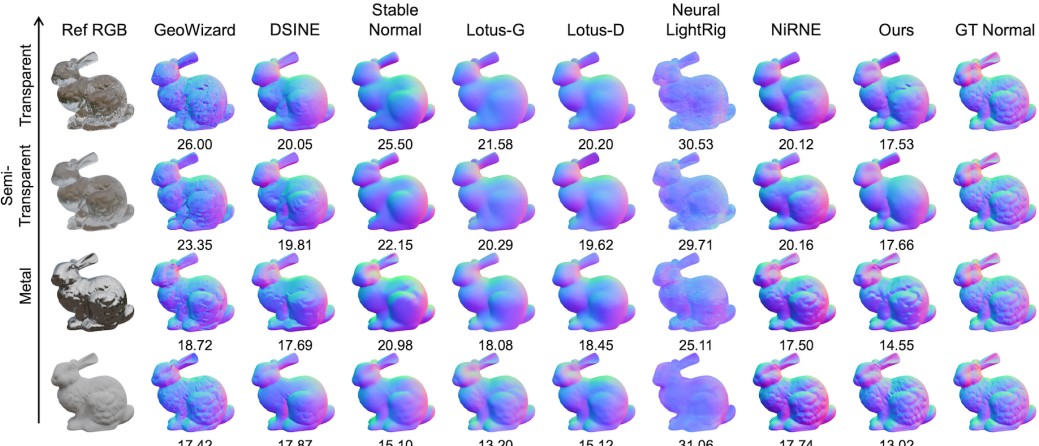

Figure 29: Analysis of each method's performance across different BRDF settings on BUNNY.

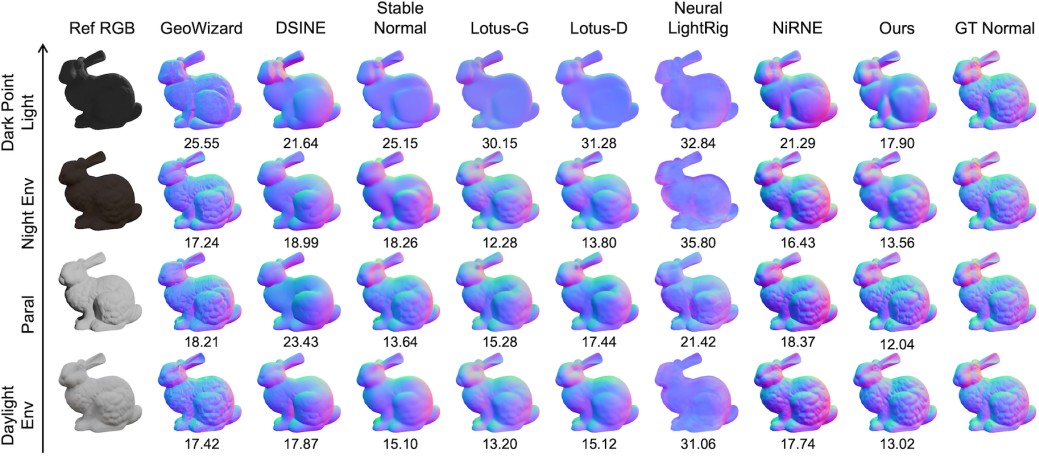

Figure 30: Analysis of each method's performance across different light settings on BUNNY, where "Paral" indicate parallel lights, "Env" stands for environment light.

# H ADDITIONAL DISCUSSION

## H.1 ANALYSIS ON LATENT'S DISTRIBUTION

We analyse the latent representations of RGB images, corresponding normal maps, and shading maps from the MultiShade after encoding them with the Stable Diffusion (Rombach et al., 2022) VAE, in order to examine whether shading maps lie closer to RGB images than the normal map in latent space. Specifically, we randomly sample 1000 objects from the dataset. For each object, we randomly select one RGB image under a random viewpoint and lighting condition, retrieve its corresponding normal map, and randomly choose one shading map computed under 9 ring lights setup, giving us 3000 samples in total. Each sample has size $72 \times 72 \times 8$. We average each latent across the $72 \times 72$ spatial dimensions and apply t-SNE to obtain a 3D embedding for each sample. This results in three sets of $1000 \times 3$ vectors, which we visualize in Fig. 31. As shown, the shading latents exhibit a tendency to cluster closer to RGB image latents.

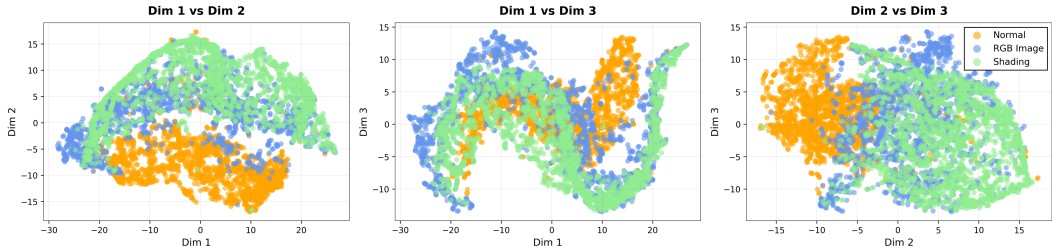

Figure 31: Visualization of the distribution of the downsampled latents of normal maps, RGB images, and shading maps from MultiShade, shown from three orthogonally projected views.

## H.2 DISCUSSION ON SHADING SEQUENCE'S ROBUSTNESS

We conduct a 5-run experiment to evaluate how perturbations in the shading sequence affect the final normal estimation. In each run, identical Gaussian noise of varying magnitudes is added to one or multiple shading maps and the surface normal. The results on BUNNY from LUCES (Mecca et al., 2021) show that the shading sequence is noticeably robust to the perturbations. When noise is injected into a single shading map, the change in mean angular error over 5 runs remains small ($1.537°$ to $5.233°$), and is substantially lower than perturbing the normal map directly ($3.235°$ to $24.421°$), even under the strongest disturbance. A larger deviation emerges only when 9 shading maps are perturbed simultaneously ($3.612°$ to $20.713°$), yet the shading-sequence formulation still remains more robust than directly perturbing normals. These observations confirm that predicting normals from shading sequences provides an inherent degree of robustness to noise.

Table 12: Quantitative analysis of how perturbations to shading sequences and normal maps affect the mean angular error relative to the clean version. 'x shading maps' indicates that x frames in the shading sequence are perturbed by Gaussian noise. The first row specifies the noise magnitude.

| Perturbed Frames | 0.050 | 0.100 | 0.200 | 0.300 | 0.400 |
|---|---|---|---|---|---|
| 1 Shading Map | 1.537 | 2.405 | 3.479 | 4.452 | 5.233 |
| 5 Shading Maps | 2.907 | 4.766 | 7.514 | 10.868 | 14.033 |
| 9 Shading Maps | 3.612 | 5.745 | 10.468 | 15.727 | 20.713 |
| Normal Map | 3.235 | 6.454 | 12.764 | 18.825 | 24.421 |

## H.3 DISCUSSION ON DIFFERENT SHADING-BASED PARADIGM

To compare the two paradigms, *i.e*, (1) first estimating normals and then computing a shading loss, and (2) our proposed paradigm that directly predicts the shading sequence and then derives normals, we evaluate the effect of incorporating a rendering-based shading loss into an image diffusion model. Specifically, we apply the shading loss to Lotus-G (He et al., 2024a), where the loss is defined

as the L1 distance between the predicted shading sequence and the ground truth sequence. Each shading map is computed as the dot product between the predicted surface normal and the preset ring-light path, with negative values clamped to 0. This variant ("Lotus w/ shad.") is trained on the MultiShade dataset. The results show that Lotus w/ shad achieves improved performance (16.20° MAE on the LUCES benchmark). However, it still falls short of our method (14.48°). This supports the effectiveness of our framework: estimating the entire shading sequence with a video diffusion model allows the network to capture the interrelationships among shading maps that contain rich geometric information, thereby improving the quality of the estimated shading sequence and enabling more accurate normal estimation.

Table 13: Comparison between "Lotus w/ shad." and the proposed RoSE on the LUCES benchmark. Numbers indicate MAE ($\downarrow$) in degrees. **Bold** numbers indicate the best results.

| Method | BALL | BELL | BOWL | BUDDHA | BUNNY | CUP | DIE | HIPPO | HOUSE | JAR | OWL | QUEEN | SQUIRREL | TOOL | Mean |
|---|---|---|---|---|---|---|---|---|---|---|---|---|---|---|---|
| Lotus w/ shad. | 19.65 | 7.66 | 9.76 | **14.95** | 9.43 | 20.45 | 12.43 | **11.00** | 36.54 | 13.06 | 23.16 | **18.01** | 18.07 | 12.64 | 16.20 |
| Ours | **9.09** | **5.94** | **6.84** | 17.58 | **12.70** | **13.80** | **8.26** | 14.14 | 36.79 | **5.93** | **19.60** | 19.99 | **21.34** | **10.66** | **14.48** |

### H.4 COMPARISON WITH MULTI-VIEW NORMAL ESTIMATION METHODS

We have conducted an additional comparison with multi-view normal estimation methods, including Era3D (Li et al., 2024a) and Unique3D Wu et al. (2024), results are shown in Table 14. On the LUCES benchmark, Era3D and Unique3D achieve mean angular errors of 43.33° and 23.00°, respectively, both substantially worse than the proposed RoSE. Particularly, although Unique3D produces visually high-contrast normal maps, the underlying geometric details are inaccurate, and its performance degrades significantly on objects with complex shadow patterns, which is consistent with our observations of 3D misalignment.

Table 14: Quantitative comparison with multi-view normal estimation methods on the LUCES benchmark. **Bold** number indicates the best results.

| Method | BALL | BELL | BOWL | BUDDHA | BUNNY | CUP | DIE | HIPPO | HOUSE | JAR | OWL | QUEEN | SQUIRREL | TOOL | Mean |
|---|---|---|---|---|---|---|---|---|---|---|---|---|---|---|---|
| Era3D | 73.02 | 29.53 | 90.85 | 27.85 | 22.82 | 67.50 | 49.73 | 28.32 | 48.87 | 38.92 | 27.86 | 32.24 | 32.13 | 37.03 | 43.33 |
| Unique3D | 14.89 | 13.52 | 13.76 | 24.68 | 16.00 | 34.76 | 19.10 | 16.52 | 44.74 | 15.80 | 28.66 | 26.02 | 26.01 | 27.48 | 23.00 |
| **Ours** | **9.09** | **5.94** | **6.84** | **17.58** | **12.70** | **13.80** | **8.26** | **14.14** | **36.79** | **5.93** | **19.60** | **19.99** | **21.34** | **10.66** | **14.48** |

### H.5 ANALYSIS ON 3D RECONSTRUCTION

We further evaluate the quality of the reconstructed surfaces obtained from both the predicted and ground truth normal maps on the DiLiGenT and LUCES datasets, using the reconstruction method of (Cao et al., 2021). The results are reported in Table 15 and Table 16. By measuring the RMSE (Cao et al., 2021) between the reconstructed and groundtruth surfaces (obtained by reconstruction from the groundtruth normal map), we observe that our method consistently achieves state-of-the-art performance across both benchmarks. These results demonstrate that RoSE not only produces accurate normal maps but also preserves high-fidelity geometric structures after surface reconstruction, further validating the effectiveness of the proposed method.

Table 15: Quantitative comparison of surface normal RMSE ($\downarrow$) on the DiLiGenT dataset. Results reported with values $\times 10$. **Bold** number indicates the best results.

| Method | BALL | BEAR | BUDDHA | CAT | COW | GOBLET | HARVEST | POT1 | POT2 | READING | AVG |
|---|---|---|---|---|---|---|---|---|---|---|---|
| Lotus-G | 1.08 | 1.31 | 0.55 | 1.19 | 1.58 | 0.84 | 1.83 | 1.96 | 0.84 | 1.18 | 1.24 |
| Lotus-D | 2.37 | **0.63** | 0.52 | 1.02 | 1.30 | **0.76** | 2.25 | 1.91 | 1.01 | **1.02** | 1.28 |
| Neural LightRig | 1.07 | 0.88 | 1.51 | 2.69 | 1.35 | 1.26 | 5.75 | 2.40 | 0.91 | 1.38 | 1.92 |
| NiRNE | 1.46 | 0.94 | 0.48 | **0.77** | 1.47 | 2.12 | 1.54 | **0.98** | 0.88 | 1.63 | 1.23 |
| **Ours** | **0.79** | 0.88 | **0.33** | 0.92 | **0.65** | 1.16 | **1.24** | 1.35 | **0.75** | 1.33 | **0.95** |

Table 16: Quantitative comparison of surface normal RMSE (↓) on the LUCES dataset. Results reported with values ×10. **Bold** number indicates the best results.

| Method | BALL | BELL | BOWL | BUDDHA | BUNNY | CUP | DIE | HIPPO | HOUSE | JAR | OWL | QUEEN | SQUIRREL | TOOL | AVG |
|---|---|---|---|---|---|---|---|---|---|---|---|---|---|---|---|
| Lotus-G | 0.89 | 0.36 | 1.33 | 1.12 | 0.72 | 1.18 | 0.67 | 1.23 | 1.23 | 0.57 | **0.83** | 0.55 | 1.30 | 0.34 | 0.88 |
| Lotus-D | 0.89 | 0.38 | 1.09 | 1.13 | 0.73 | 1.82 | 0.75 | 1.09 | 1.54 | 0.56 | 0.89 | 0.46 | 0.86 | 0.31 | 0.89 |
| Neural LightRig | **0.31** | 0.51 | 3.02 | **0.57** | 0.69 | 0.98 | 1.71 | 1.51 | 1.28 | 1.88 | 1.05 | 0.45 | 0.60 | 0.65 | 1.09 |
| NiRNE | 0.77 | 0.76 | 2.12 | 0.70 | 0.64 | 1.10 | 1.02 | 1.71 | 1.12 | 0.75 | 1.93 | 0.75 | 0.71 | 0.71 | 1.05 |
| **Ours** | 0.34 | **0.19** | **0.80** | 0.69 | **0.53** | **0.72** | **0.54** | **0.79** | **0.67** | **0.42** | 1.22 | **0.25** | **0.54** | **0.22** | **0.57** |

## H.6 PERFORMANCE ON NORMAL ESTIMATION USING VIDEO DIFFUSION MODEL

We have conducted an additional comparison with the variant that using SV3D to directly predict the single-frame surface normal (noted as "SVD-nml"), the results are shown in Table 17. The average result on LUCES benchmark dataset is $20.61°$, indicating that the dense information geometrically encoded in the video model does not play a critical role in this setting, and the model also loses its ability to make use of temporal information.

Table 17: Quantitative comparison with "SVD-nml" on the LUCES benchmark. **Bold** number indicates the best results.

| Method | BALL | BELL | BOWL | BUDDHA | BUNNY | CUP | DIE | HIPPO | HOUSE | JAR | OWL | QUEEN | SQUIRREL | TOOL | Mean |
|---|---|---|---|---|---|---|---|---|---|---|---|---|---|---|---|
| SVD-nml | 17.20 | 9.96 | 19.69 | 22.28 | 14.77 | 18.32 | 11.47 | 18.23 | 49.74 | 10.49 | 26.95 | 25.04 | 26.03 | 18.29 | 20.61 |
| **Ours** | **9.09** | **5.94** | **6.84** | **17.58** | **12.70** | **13.80** | **8.26** | **14.14** | **36.79** | **5.93** | **19.60** | **19.99** | **21.34** | **10.66** | **14.48** |