# OpenReview forum: "Monocular Normal Estimation via Shading Sequence Estimation"
_ICLR.cc/2026/Conference — ICLR 2026 Oral_

### Official Review · Reviewer_fSqW · 2025-10-21

**Soundness:** 4
**Presentation:** 3
**Contribution:** 4
**Rating:** 6
**Confidence:** 4

**Summary:**

This paper targets monocular normal map estimation, which aims to recover surface normal information from a single RGB object image under arbitrary lighting conditions. The core technical approach is to formulate the normal estimation problem as a shading sequence estimation task, using an image-to-video generative model to predict a sequence of shading images under predefined parallel light directions from a grayscale input image. This sequence is then converted into a normal map using an ordinary least square solver. The main contributions are to introduce the shading sequence as a more sensitive geometric representation to address 3D misalignment issues. Employing a video diffusion model to generate consistent shading sequences is also interesting. A new synthetic dataset MultiShade is also proposed. Experimental results demonstrate that RoSE achieves state-of-the-art performance on real-world benchmark datasets such as DiLiGenT and LUCES, while also excels on the synthetic MultiShade dataset.

**Strengths:**

The paper has some clear strengths:
Transforming normal estimation into a shading sequence generation task and leveraging video generative models to handle temporal consistency are very interesting choice. This introduces a new perspective for normal reconstruction.

The paper uses shading sequences as the training target, which is more sensitive to geometric variations than directly predicting normal maps.

The method achieves state-of-the-art performance on real-world benchmark datasets.

**Weaknesses:**

The key idea and technical module, which uses a video generation model to predict shading sequences, is similar to NormalCrafter [Bin et al 2025]. However, it seems that it is not discussed in this paper.

I'm not sure if using grayscale input images is a good idea. Despite being claimed in the paper that it can "eliminating redundant chromatic information", but did you conduct any ablation studies on this? For complex materials (e.g. metallic reflections), color may provide additional geometric hints, potentially leading to information loss.

The key claim of “reducing 3D misalignment” is only supported by qualitative comparisons and MAE metrics, without quantifying 3D reconstruction errors. If we use RoSE to estimate multi-view normal maps of a single object, can we achieve better consistency and reconstruction accuracy?

Estimating monocular normals using dense video models consumes excessive additional resources. At this point, I'm still uncertain whether such overhead would be widely accepted in practical applications.

The paper claims directly estimating single normal maps will lead to 3D inconsistency. I think multi-view normal estimation may alleviate this problem to some extent. The representative works such as Unique3D and Era3D also show detailed normal estimation results via a multi-view setting, but they are not compared and discussed in the paper.

**Questions:**

Will Negative-clamping introduce nonlinearity, which may violate the OLS linearity assumption?  It is not a perfect linear equivalent.

The choice of latitude for the ring light is described as an “appropriate choice” without specifying exact values or a computational formula? Is ring light setup the best choice? How do we "ensuring that each point is illuminated by at least three sources with positive shading values"?

Just curious: How can we verify that the improvement in normal estimation accuracy comes from estimating shading, rather than the extremely dense information provided by the video model? Is it possible to achieve similar results by estimating depth or directly estimating normals instead?

---

> ### Author Response · Authors · 2025-11-21
> **Response to Reviewer fSqW**
>
> **Q1: Similarity with NormalCrafter.**
>
> Although both methods employ similar backbone models (i.e., Stable Video Diffusion Model, SVD), NormalCrafter and our method differ fundamentally in their objectives. NormalCrafter uses SVD to predict normal frames frame-by-frame and focuses primarily on enforcing temporal consistency for video-based normal estimation. In contrast, our method leverages SVD for monocular normal estimation by predicting shading sequence, emphasising achieving high estimation accuracy across diverse lighting conditions and material properties.
>
> We have added a related discussion to **Sec. 2**, with the revision shown below:
>
> > ***Video Generation.** … In 3D estimation, recent work (NormalCrafter) employs a video diffusion model for normal estimation, but focuses on predicting per-frame normals for an input video. In contrast, our work leverages the capability of video generative models to predict a shading sequence that follows a pre-defined light path under multiple parallel light sources, using only a single input image. This enables monocular normal estimation for objects with diverse shapes and materials.*
> >
>
> **Q2: Effectiveness of the grayscale input.**
>
> We have conducted an ablation study that replaces the grayscale input with an RGB input during training and inference. The performance drops 0.79° on the LUCES benchmark, indicating the importance of grayscale input to eliminate redundant chromatic information for shading sequence prediction.
>
> This ablation study has been included in **Sec. 4.4** with revision shown below:
>
> > *We retrain a variant of RoSE that replaces the grayscale input with an RGB input (i.e., Ours w/ RGB). The performance drops 0.79° on the LUCES benchmark. For metallic objects (i.e., Bell and Cup), we do not observe a clear advantage when using RGB input. These results indicate the importance of the grayscale input, which removes redundant chromatic information and enables more accurate shading sequence estimation.*
> >

---

> ### Author Response · Authors · 2025-11-21
>
> **Q3: lacking qualification in 3D reconstruction errors.**
>
> We have evaluated the 3D reconstruction quality using RMSE [C]. The SOTA performance on both DiLiGenT and LUCES shows that the proposed RoSE indeed results in a more accurate reconstructed surface.
>
> The results are reported in **Sec. I.5** in the appendix.
>
> > *We further evaluate the quality of the reconstructed surfaces obtained from the estimated normal maps on the DiLiGenT and LUCES datasets, using the reconstruction method of [C]. The results are reported in Tab. 15 and Tab. 16. By measuring the RMSE [C] between the reconstructed and groundtruth surfaces (obtained by reconstruction from the groundtruth normal map), we observe that our method consistently achieves state-of-the-art performance across both benchmarks. These results demonstrate that RoSE not only produces accurate normal maps but also preserves high-fidelity geometric structures after surface reconstruction, further validating the effectiveness of the proposed method.*
> >
>
> **Table 15**: Quantitative comparison of surface normal RMSE on the DiLiGenT dataset. Results reported with values x 10. **Bold** number indicates the best results.
>
> |  | **Ball** | **Bear** | **Buddha** | **Cat** | **Cow** | **Goblet** | **Harvest** | **Pot1** | **Pot2** | **Reading** | **AVG** |
> | --- | --- | --- | --- | --- | --- | --- | --- | --- | --- | --- | --- |
> | Lotus-G | 1.08  | 1.31  | 0.55  | 1.19  | 1.58  | 0.84  | 1.83  | 1.96  | 0.84  | 1.18  | 1.24  |
> | Lotus-D | 2.37  | **0.63**  | 0.52  | 1.02  | 1.30  | **0.76**  | 2.25  | 1.91  | 1.01  | **1.02**  | 1.28  |
> | Neural LightRig | 1.07  | 0.88  | 1.51  | 2.69  | 1.35  | 1.26  | 5.75  | 2.40  | 0.91  | 1.38  | 1.92  |
> | NiRNE | 1.46  | 0.94  | 0.48  | **0.77**  | 1.47  | 2.12  | 1.54  | **0.98**  | 0.88  | 1.63  | 1.23  |
> | **Ours** | **0.79**  | 0.88  | **0.33**  | 0.92  | **0.65**  | 1.16  | **1.24**  | 1.35  | **0.75**  | 1.33  | **0.95**  |
>
> **Table 16**: Quantitative comparison of surface normal RMSE on the LUCES dataset. Results reported with values x 10. **Bold** number indicates the best results.
>
> | Method | Ball | Bell | Bowl | Buddha | Bunny | Cup | Die | Hippo | House | Jar | Owl | Queen | Squirrel | Tool | AVG |
> | --- | --- | --- | --- | --- | --- | --- | --- | --- | --- | --- | --- | --- | --- | --- | --- |
> | Lotus-G | 0.89 | 0.36 | 1.33 | 1.12 | 0.72 | 1.18 | 0.67 | 1.23 | 1.23 | 0.57 | **0.83** | 0.55 | 1.30 | 0.34 | 0.88 |
> | Lotus-D | 0.89 | 0.38 | 1.09 | 1.13 | 0.73 | 1.82 | 0.75 | 1.09 | 1.54 | 0.56 | 0.89 | 0.46 | 0.86 | 0.31 | 0.89 |
> | Neural LightRig | **0.31** | 0.51 | 3.02 | **0.57** | 0.69 | 0.98 | 1.71 | 1.51 | 1.28 | 1.88 | 1.05 | 0.45 | 0.60 | 0.65 | 1.09 |
> | NiRNE | 0.77 | 0.76 | 2.12 | 0.70 | 0.64 | 1.10 | 1.02 | 1.71 | 1.12 | 0.75 | 1.93 | 0.75 | 0.71 | 0.71 | 1.05 |
> | **Ours** | 0.34 | **0.19** | **0.80** | 0.69 | **0.53** | **0.72** | **0.54** | **0.79** | **0.67** | **0.42** | 1.22 | **0.25** | **0.54** | **0.22** | **0.57** |
>
> [C]  Cao, Xu, et al. "Normal integration via inverse plane fitting with minimum point-to-plane distance." Proceedings of the IEEE/CVF Conference on Computer Vision and Pattern Recognition. 2021.
>
> **Q4: Can RoSE achieve better consistency and reconstruction accuracy by predicting multi-view normal maps?**
>
> RoSE cannot obtain performance gains under a multiview setup. This is because information across different views is not shared within the RoSE framework, making the multiview setting essentially equivalent to independently predicting normals for each view.
>
> **Q5: Would computational overhead be widely accepted in practical applications?**
>
> We report RoSE’s inference speed for single-image normal estimation and compare it with existing methods (see **Tab. 7** in the appendix). Although RoSE is slower than methods such as NiRNE, it remains faster than Neural LightRig and GeoWizard. Importantly, RoSE achieves the top two performances across all benchmarks and ranks first in average accuracy in normal estimation.  Moreover, our method is well-suited for applications that do not require real-time performance, such as serving as an auxiliary signal for 3D generation or offline normal map estimation.
>
> We have included this discussion in **Sec.D** in the appendix:
>
> > **_Testing details._** … *All testing processes are performed on a single RTX A6000 Ada GPU. The total runtime includes the video diffusion model inference with 25 denoising steps and the shading to normal computation, the latter adding only a negligible cost of 0.045 seconds per object. For completeness, we also report the inference time of other methods for reference.*
> >
>
> **Table 7**: Average inference time of monocular normal estimation methods per image (in seconds).
>
> | **Method** | **GeoWizard** | **DSINE** | **StableNormal** | **Lotus-G** | **Lotus-D** | **Neural LightRig** | **NiRNE** | **Ours** |
> | --- | --- | --- | --- | --- | --- | --- | --- | --- |
> | Time | 101.11 | 0.83 | 1.52 | 0.61 | 0.59 | 93.73 | 0.31 | 10.57 |

---

> ### Author Response · Authors · 2025-11-21
>
> **Q6: Comparison with the state-of-the-art multi-view normal estimator**
>
> We have conducted an additional comparison with multi-view normal estimation methods, including Unique3D and Era3D. On the LUCES benchmark, Era3D and Unique3D achieve mean angular errors of **43.33°** and **23.00°**, respectively, both substantially worse than the proposed RoSE.
>
> We have added this experiment to **Sec. I.4** in the appendix, with the revision shown below:
>
> > ***I.4 Comparison with Multi-view Normal Estimation Methods**
> We have conducted an additional comparison with multi-view normal estimation methods, including Unique3D and Era3D, results are shown in Tab. 14. On the LUCES benchmark, Era3D and Unique3D achieve mean angular errors of 43.33° and 23.00°, respectively, both substantially worse than the proposed RoSE. Particularly, although Unique3D produces visually high-contrast normal maps, the underlying geometric details are inaccurate, and its performance degrades significantly on objects with complex shadow patterns, which is consistent with our observations of 3D misalignment.*
> >
>
> **Table 14**: Quantitative comparison with multi-view normal estimation methods on the LUCES benchmark. **Bold** numbers indicate the best results.
>
> |  | **Ball** | **Bell** | **Bowl** | **Buddha** | **Bunny** | **Cup** | **Die** | **Hippo** | **House** | **Jar** | **Owl** | **Queen** | **Squirrel** | **Tool** | **AVG** |
> | --- | --- | --- | --- | --- | --- | --- | --- | --- | --- | --- | --- | --- | --- | --- | --- |
> | Era3D | 73.02 | 29.53 | 90.85 | 27.85 | 22.82 | 67.50 | 49.73 | 28.32 | 48.87 | 38.92 | 27.86 | 32.24 | 32.13 | 37.03 | 43.33 |
> | Unique3D | 14.89 | 13.52 | 13.76 | 24.68 | 16.00 | 34.76 | 19.10 | 16.52 | 44.74 | 15.80 | 28.66 | 26.02 | 26.01 | 27.48 | 23.00 |
> | Ours | **9.09** | **5.94** | **6.84** | **17.58** | **12.70** | **13.80** | **8.26** | **14.14** | **36.79** | **5.93** | **19.60** | **19.99** | **21.34** | **10.66** | **14.48** |
>
>
> **Q7: Will negative-clamping introduce nonlinearity, which may violate the OLS linearity assumption?**
>
> Negative clamping does not affect the OLS result when it is applied only to points with positive shading values. We have added a clarification near **Eq. (4)** in **Sec. 3.1**.
>
> > *The solution is unique when L is full rank. In practice, the introduction of the max(.,0) operation would introduce bias if we directly apply OLS to the estimated shading sequence. To mitigate this issue, we treat only shading values greater than 0 as valid equations for OLS when solving the normal.*
> >
>
> **Q8: Chosen latitude and validation on the light setup.**
>
> We use a latitude of 45° to maintain a balanced lighting pattern. Details have been added to **Sec. 3.2**.
>
> Finding an optimal configuration is challenging due to the large number of possible choices and the very high computational cost. We chose the ring light setup because it is simple and easier for the model to learn. A complex light path may degrade performance. For example, we compare with the spiral setup, where the elevation decreases from 60° to 30° while rotating around the z-axis, and the performance is worse than the 45° configuration (**17.60°**). We believe this is because the 45° setup provides the most balanced brightness, which the model can fit more effectively.  This discussion has been added to **Sec. 4.4**.
>
> The revised **Sec. 3.2** and **Sec. 4.4** is shown below.
>
> > **_Sec. 3.2._** _… With an appropriate choice of the ring’s latitude (45° in our setup)_
> >
> >**_Sec. 4.4. …_** _We train RoSE using an alternative light path in which the elevation gradually decreases from 60° to 30° while rotating 360° around the z-axis. We denote this variant as ``Ours w/ spiral’’. As shown in Tab. 6, this more complex light path leads to a performance drop (MAE of 17.60°). This result highlights that the proposed ring-light setup is an effective and efficient design for the model to predict the shading sequence._
> >
> **Q9: How can we ensure that at least three sources illuminate each point?**
>
> We verify that each surface point receives at least three effective light sources from both theoretical and empirical perspectives.
>
> **Theoretically**, we adopt a ring light configuration, which ensures a balanced light distribution in the shading sequence as described in **Lemma 1**. Particularly, 9 lights in the ring setup are sufficient to guarantee at least three positive shading values for each point.
>
> **Empirically**, we analyse the shading sequences of the test objects estimated by RoSE and count, for each surface point, how many shading maps yield positive shading. The results show that each object consistently satisfies the condition of having at least 3 positive shading values per point, which, in turn, supports a valid OLS solution.

---

> ### Author Response · Authors · 2025-11-21
>
> **Q10: Performance in normal estimation when using a video model as the base model.**
>
> We replace the shading sequence prediction objective with single-frame normal estimation and evaluate the performance on the LUCES benchmark. The average result is **20.61°** (worse than RoSE, **14.48°**), indicating that the dense information geometrically encoded in the video model does not play a critical role in this setting, and the model also loses its ability to make use of temporal information. We believe that a video diffusion model functions most effectively as an image sequence predictor. For monocular normal estimation, this naturally aligns with our goal of predicting a shading sequence, which also forms one of the key contributions of this work.
>
> This discussion has been included in the **Sec. I.6** in the appendix.
>
> > *We have conducted an additional comparison with the variant that using SV3D to directly predict the single-frame surface normal (noted as “SVD-nml”), the results are shown in **Tab. 17**. The average result on LUCES benchmark dataset is **20.61*°***, indicating that the dense information geometrically encoded in the video model does not play a critical role in this setting, and the model also loses its ability to make use of temporal information.*
> >
>
> **Table 17:** Quantitative comparison with ``SVD-nml'' on the LUCES benchmark. **Bold** numbers indicate the best results.
>
> | Method | Ball | Bell | Bowl | Buddha | Bunny | Cup | Die | Hippo | House | Jar | Owl | Queen | Squirrel | Tool | Mean |
> | --- | --- | --- | --- | --- | --- | --- | --- | --- | --- | --- | --- | --- | --- | --- | --- |
> | SVD-nml | 17.20 | 9.96 | 19.69 | 22.28 | 14.77 | 18.32 | 11.47 | 18.23 | 49.74 | 10.49 | 26.95 | 25.04 | 26.03 | 18.29 | 20.61 |
> | **Ours** | **9.09** | **5.94** | **6.84** | **17.58** | **12.70** | **13.80** | **8.26** | **14.14** | **36.79** | **5.93** | **19.60** | **19.99** | **21.34** | **10.66** | **14.48** |

---

### Official Review · Reviewer_81qY · 2025-10-27

**Soundness:** 3
**Presentation:** 3
**Contribution:** 3
**Rating:** 8
**Confidence:** 5

**Summary:**

The paper proposes RoSE, a monocular surface normal estimator that reformulates normal estimation as “shading sequence estimation.” Instead of directly predicting a normal map from a single RGB image, RoSE leverages an image-to-video diffusion model to generate a short sequence of grayscale, lambertian shadings of the input object under a predefined set of parallel lights; the final normals are then recovered analytically via Ordinary Least Squares (OLS). This change of target is motivated by the claim that shading sequences are more sensitive to geometric variations than the compact color encoding of normal maps, reducing “3D misalignment” (over-smooth or geometrically inconsistent normals). To train and evaluate the model, the paper also curate a synthetic dataset MultiShade with diverse shapes, materials, and lighting. Experiments on DiLiGenT, LUCES, and the proposed MultiShade dataset show state-of-the-art performance among monocular normal estimation methods.

**Strengths:**

- The paper proposes an interesting approach that combines the generative capabilities of video models with classic shape-from-shading priors to tackle geometric reconstruction.

- Formulating the synthesis of shading images directly as a video generation problem—rather than relying primarily on image generation as in prior work is an interesting point. This also aligns with recent findings suggesting that video generators already possess strong general-purpose visual understanding and synthesis abilities [1].

- The experimental results appear strong and promising.

**Weaknesses:**

- The generated Shading Sequence is central to the method, yet the experiments lack both qualitative and quantitative evaluations of this component (see Questions below).

- The discussion of output variation (L357) and the failure cases (last section of the appendix) is too generic. At L357, the inconsistencies across cases are attributed to “the inherent variance of the model,” which is uninformative and unsatisfying. More concrete analysis is encouraged: for example, examining how input illumination, texture complexity, and BRDF properties influence the recovered normal accuracy.

- The paper would benefit from additional visualizations, especially examples of the generated Shading Sequence and relighting videos.

**Questions:**

- The closed-form OLS solution for normal map estimation requires reasonably accurate shading sequences. How robust is the generated Shading Sequence in practice? If there is bias in the estimated shading sequence from the network, to what extent does it degrade the final normal quality?

- A classic way to incorporate shape-from-shading into deep learning is via a render-loss (dating back at least to [2]), where predicted normals/SVBRDF are rendered under multiple point lights and gradients are backpropagated. When training on synthetic datasets, it is possible to guarantee that the lighting condition used for render loss also exactly align with the proposed lighting setup in this paper. Hence an interesting question is: how do the trade-offs compare between (a) training with a render-loss, i.e., the network predicts normals directly but the loss is render (shading)-aware during training), and (b) the proposed approach in this paper (first generate the shading, then solve for normals offline)?

[1] Wiedemer, Thaddäus, et al. “Video models are zero-shot learners and reasoners.” arXiv:2509.20328 (2025).

[2] Deschaintre, Valentin, et al. “Single-image SVBRDF capture with a rendering-aware deep network.” ACM TOG 37.4 (2018): 1–15.

---

> ### Author Response · Authors · 2025-11-21
> **Response to Reviewer 81qY**
>
> **Q1: Qualitative and quantitative evaluations of shading sequence; more visualization.**
>
> We have conducted both qualitative and quantitative analyses of the predicted shading sequences on the DiLiGenT and LUCES datasets to illustrate RoSE’s ability to recover accurate shading sequences. Quantitative results are added to **Sec. 4.2, Tab. 4.** while qualitative results are added to **Sec. G.2.** We have also uploaded the corresponding videos to the anonymous link in the appendix.
>
> > ***Sec 4.2. Analysis on Shading Sequence**. We conduct quantitative analyses of the predicted shading sequences on the LUCES dataset to illustrate RoSE’s ability to recover accurate shading sequences. For each predicted frame, the shading of all other methods (including the ground truth) is computed as the dot product between the ground truth light direction and the surface normal, with negative values clamped to zero. We use PSNR ($\uparrow$), SSIM ($\uparrow$), and LPIPS ($\downarrow$) as the evaluation metrics, as shown in Tab.4. The results demonstrate that RoSE achieves SOTA quality in predicted shading sequence, which also align with the results of normal estimation.*
> >
>
> **Tab. 4**: Quantitative comparison on estimated shading sequence in terms of PSNR ($\uparrow$), SSIM ($\uparrow$), and LPIPS ($\downarrow$) on LUCES benchmark dataset. **Bold** number indicates the best results.
>
> | **Metrics** | **GeoWizard** | **DSINE** | **StableNormal** | **Lotus-G** | **Lotus-D** | **Neural LightRig** | **NiRNE** | **Ours** |
> | --- | --- | --- | --- | --- | --- | --- | --- | --- |
> | PSNR | 16.86 | 17.05 | 18.40 | 19.19 | 18.80 | 17.88 | 18.99 | **20.74** |
> | SSIM | 0.69 | 0.72 | 0.74 | 0.76 | 0.75 | 0.71 | 0.75 | **0.77** |
> | LPIPS | 0.28 | 0.31 | 0.30 | 0.27 | 0.29 | 0.28 | 0.27 | **0.26** |
>
> **Q2: Discussion of the failure cases is too generic.**
>
> We conduct a detailed analysis across variations in lighting, textures, and BRDFs through three groups of controlled experiments. The first group examines how different lighting conditions, including simple light, parallel light, point light, and challenging light, influence performance, with texture and BRDF fixed. The second group evaluates the impact of texture complexity by comparing a simple texture with three spatially varying textures, while holding lighting and BRDF constant. The third group isolates material effects by varying the BRDF under simple lighting and texture settings. The results (see **Fig. 28, Fig. 29, and Fig. 30**) show that our method consistently produces high-quality normal maps relative to other methods. However, performance does degrade under highly complex textures, challenging lighting conditions, or difficult material properties.
>
> Please refer to **Sec. H** for more details.

---

> ### Author Response · Authors · 2025-11-21
>
> **Q3: Robustness analysis of the generated shading sequence.**
>
> As the quality of the estimated shading sequence directly determines the quality of the estimated normals, large deviations in the shading sequence would naturally lead to errors in the computed normal maps. However, because the normal map is inferred from the entire shading sequence, inaccuracies in a portion of it do not substantially affect overall performance. To further verify this, we conduct an experiment that examines the extent to which perturbations in the shading sequence affect the final normal estimation results. Detailed analyses are provided in the **Sec. I.2** in the appendix.
>
> > **_I.2 Discussion on Shading Sequence’s Robustness_**.
> *We conduct a 5-run experiment to evaluate how perturbations in the shading sequence affect the final normal estimation. In each run, identical Gaussian noise of varying magnitudes is added to one or multiple shading maps and the surface normal. The results in Tab. 12 on Bunny from LUCES show that the shading sequence is noticeably robust to the perturbations. When noise is injected into a single shading map, the change in mean angular error over 5 runs remains small (**1.537*° *to 5.233*°***), and is substantially lower than perturbing the normal map directly (**3.235*° *to 24.421*°***), even under the strongest disturbance. A larger deviation emerges only when nine shading maps are perturbed simultaneously (**3.612*° *to 20.713*°***), yet the shading-sequence formulation still remains more robust than directly perturbing normals. These observation confirm that predicting normals from shading sequences provides an inherent degree of robustness to noise.*
> >
> **Table 12**: Quantitative analysis of how perturbations to shading sequences and normal maps affect the mean angular error relative to the clean version. `x shading maps' indicates that x frames in the shading sequence are perturbed by Gaussian noise. The first row specifies the noise magnitude.
>
> | Perturbed Frames | **0.050** | **0.100** | **0.200** | **0.300** | **0.400** |
> | --- | --- | --- | --- | --- | --- |
> | 1 Shading Map | 1.537 | 2.405 | 3.479 | 4.452 | 5.233 |
> | 5 Shading Maps | 2.907 | 4.766 | 7.514 | 10.868 | 14.033 |
> | 9 Shading Maps | 3.612 | 5.745 | 10.468 | 15.727 | 20.713 |
> | Normal Map | 3.235 | 6.454 | 12.764 | 18.825 | 24.421 |
>
> **Q4: Comparison with render-loss.**
>
> We evaluate the effect of the rendering shading loss by applying it to an image-based diffusion model, Lotus G, to predict normals. The results show that supervising with the shading sequence brings improvement, reaching **16.20**° on the LUCES benchmark, yet it still performs below our method (**14.48**°). This validates the effectiveness of the proposed framework, which estimates the entire shading sequence using a video diffusion model that implicitly captures the interrelationships among different shading maps, thereby improving the quality of the estimated shading sequence and enabling more accurate normal estimation.
>
> The above discussion has been added to **Sec. I.3** in the appendix.
>
> > *To compare the two paradigms, i.e., (1) first estimating normals and then computing a shading loss, and (2) our proposed paradigm that directly predicts the shading sequence and then derives normals, we evaluate the effect of incorporating a rendering-based shading loss into an image diffusion model. Specifically, we apply the shading loss to Lotus-G, where the loss is defined as the L1 distance between the predicted shading sequence and the ground truth sequence. Each shading map is computed as the dot product between the predicted surface normal and the preset ring-light path, with negative values clamped to 0. This variant (”Lotus w/ shad.”) is trained on the MultiShade dataset.*
> >
> >
> > *The results show that Lotus w/ shad achieve an improved performance (i.e., 16.20° MAE on the LUCES benchmark). However, it still falls short of our method (14.48°). This supports the effectiveness of our framework: estimating the entire shading sequence with a video diffusion model allows the network to capture the interrelationships among shading maps that contains rich geometric information, thereby improving the quality of the estimated shading sequence and enabling more accurate normal estimation.*
> >

---

> > ### Comment · Reviewer_81qY · 2025-11-26
> >
> > Thanks for the response. I believe most of my questions have been addressed. I would keep my current rating of 8.

---

> > > ### Author Response · Authors · 2025-11-26
> > > **Reply to Reviewer 81qY**
> > >
> > > We sincerely appreciate your positive assessment and your thoughtful suggestions. We are pleased that our response has addressed most of your questions. Thank you again for your support!

---

### Official Review · Reviewer_uAAj · 2025-10-27

**Soundness:** 3
**Presentation:** 3
**Contribution:** 3
**Rating:** 6
**Confidence:** 4

**Summary:**

This paper addresses the long-standing 3D misalignment issue in monocular normal estimation by introducing a novel framework, RoSE. The proposed paradigm reconceptualizes normal estimation as a shading sequence estimation problem and leverages an image-to-video generative model to predict shading sequences under predefined parallel lighting conditions. The normal map is subsequently derived through a simple ordinary least squares formulation. Extensive experiments demonstrate that RoSE achieves state-of-the-art performance on multiple real-world benchmark datasets for object-level monocular normal estimation.

**Strengths:**

Originality. The paper introduces a novel approach to monocular normal estimation by rethinking the problem in the context of shading sequence estimation. This conceptual shift is quite original, as traditional methods in normal estimation usually focus on individual lighting models or directly predicting normals from images.

Quality. The paper delivers high-quality research with state-of-the-art results on multiple real-world benchmark datasets. Achieving superior performance compared to existing methods demonstrates the technical rigor of the proposed framework. The extensive experimental evaluation suggests that the approach has been thoroughly tested and validated, which is essential in establishing the robustness of the proposed method.

Clarity. The paper appears to be clear in presenting its approach. The explanation of the framework—reconceptualizing normal estimation as a shading sequence estimation problem—is concise and understandable. The use of an ordinary least squares formulation to derive the normal map is presented as a simple final step, which helps make the approach accessible without sacrificing technical detail.

Significance. The significance of this paper lies in addressing a long-standing challenge in computer vision, monocular normal estimation under varying lighting conditions. The ability to generate accurate normal maps from monocular images has numerous applications in 3D reconstruction, robotics, AR/VR, and object recognition. By introducing a novel framework that is not heavily reliant on complex sensor setups or assumptions about lighting, the proposed work could have a substantial impact on various real-world applications where obtaining ground truth normal maps is challenging.

**Weaknesses:**

1.Insufficient training details. While the paper states that RoSE is built upon the SV3D model (Voleti et al., 2024), the specific parameter configurations, training protocols, and dataset utilization are not described. Providing these details would significantly improve reproducibility.

2.Limited ablation studies. The current ablation experiments primarily focus on dataset influence. It would be valuable to include analyses of the model components — such as varying SV3D settings or substituting alternative video diffusion models — to better assess the robustness and generality of the approach.

**Questions:**

1.What is the overall inference speed of the RoSE pipeline, including shading sequence generation and normal reconstruction?

2.How does the proposed method perform when estimating normals for transparent or translucent objects, where shading cues may be ambiguous or physically inconsistent?

---

> ### Author Response · Authors · 2025-11-21
> **Response to Reviewer uAAj**
>
> **Q1: Insufficient training details.**
>
> We have expanded the training details in the **Sec. D,** appendix, covering parameter configurations, training protocols, and dataset usage. We summarize as:
>
> (1) Parameter configurations: we reuse the pretrained SV3D weights, which contain about 1.5B parameters.
>
> (2) Training protocols: we fine-tune only the attention layers and the first convolution layer in each block, resulting in about 200M trainable parameters.
>
> (3) Dataset usage: the model is trained on 90K objects with 3.1M image normal pairs, and evaluated on a validation set of 100 objects, each with 100 image normal pairs.
>
> We appreciate any further suggestions regarding missing details and will include them in the revised version of the paper.
>
> The revised **Sec. D** in the appendix is shown below:
>
> > *… with pre-trained SV3D weights (1.5B) … and fine-tune the first convolutional layer as well as all parameters within the self-attention and cross-attention modules on the MultiShade dataset, which contains 90K objects (3.1M image–normal pairs) and evaluated on a validation set of 100 objects, each providing 100 image–normal pairs.*
> >
>
> **Q2: Limited ablation studies about the model components.**
>
> We have also trained RoSE with another video diffusion variant, namely Stable Video Diffusion XL (SVD XL, 1.5B parameters), on the MultiShade dataset. On the LUCES benchmark, this variant achieves a mean angular error of **14.58°**, showing performance similar to the SV3D-based version. These results are reported in **Sec. 4.4** of the main paper. The experiments demonstrate that our method generalizes well to SVD-based video diffusion models, even when the backbone is pretrained on general video data rather than a domain-specific object-centric dataset. In future work, we seek to support other video diffusion models, including those built on DiT-based architectures. Training such models requires substantial computational resources, which makes it challenging to complete within the rebuttal period.
>
> The revised **Section 4.4** is shown below:
>
> > ***Validation on Model Variants Impact.** We have trained RoSE using a different video diffusion backbone, namely Stable Video Diffusion XL (SVD XL), on the MultiShade dataset. We denote this variant as “Ours w/ SVD XL”. On the LUCES dataset, this model achieves performance comparable to RoSE built on SV3D (14.58*° *for “Ours w/ SVD XL” and 14.48*° *for Ours). This demonstrates that our framework generalises well even when the backbone is pretrained on large scale, general purpose video data rather than a domain-specific object-centric dataset.*
> >
>
> **Q3: Inference speed of the RoSE pipeline.**
>
> We report RoSE’s inference speed for single-image normal estimation and compare it with existing methods (see **Tab. 7** in the appendix). Although RoSE is slower than methods such as NiRNE, it remains faster than Neural LightRig and GeoWizard. Importantly, RoSE achieves the top two performances across all benchmarks and ranks first in average accuracy in normal estimation.  Moreover, our method is well-suited for applications that do not require real-time performance, such as serving as an auxiliary signal for 3D generation or offline normal map estimation.
>
> We have included this discussion in **Sec.D** in the appendix:
>
> > **Testing details.** … *All testing processes are performed on a single RTX A6000 Ada GPU. The total runtime includes the video diffusion model inference with 25 denoising steps and the shading to normal computation, the latter adding only a negligible cost of 0.045 seconds per object. For completeness, we also report the inference time of other methods for reference.*
> >
>
> **Table 7**: Average inference time of monocular normal estimation methods per image (in seconds).
>
> | **Method** | **GeoWizard** | **DSINE** | **StableNormal** | **Lotus-G** | **Lotus-D** | **Neural LightRig** | **NiRNE** | **Ours** |
> | --- | --- | --- | --- | --- | --- | --- | --- | --- |
> | Time | 101.11 | 0.83 | 1.52 | 0.61 | 0.59 | 93.73 | 0.31 | 10.57 |
>
> **Q4: Performance on semi-transparent and transparent objects.**
>
> We have evaluated our method and other methods on transparent objects (see **Sec. H** for more details) and found that although our method estimates a more accurate normal map (**17.53**° for ours vs. **20.05**° for current SOTA), all methods struggle to recover detailed structure on this highly challenging material. Estimating normal maps for semi-transparent or transparent objects is known to be difficult due to subsurface reflections, complex light interactions, and substantial appearance changes driven by lighting, which are especially problematic in monocular settings. Extending our method to handle such materials better is an important direction for future work.
>
> We have added this discussion to **Sec. 5**.

---

### Official Review · Reviewer_otsa · 2025-10-31

**Soundness:** 3
**Presentation:** 3
**Contribution:** 4
**Rating:** 6
**Confidence:** 3

**Summary:**

The paper introduces the RoSE method to solve the 3D misalignment challenge in Monocular Normal Estimation (MNE). The authors reframe the MNE task from direct normal map prediction to estimating a shading sequence. This sequence, which captures pixel brightness under controlled lighting, is considered more sensitive to fine geometric details, providing a superior training signal. ROSE is trained using the new MultiShade dataset to enhance robustness. Experiments show the method achieves state-of-the-art performance on real-world benchmarks, successfully reconstructing finer geometric details with better 3D alignment.

**Strengths:**

1. It reformulates the task as shading sequence estimation effectively addresses the core issue of 3D misalignment and oversmoothness. It is a novel new formulation.
2. The new synthetic dataset includes diverse materials, light conditions, and material augmentation, successfully improving the model's generalization ability. The new dataset is also a contribution.
3. It achieves superior performance on key real-world benchmark datasets like DiLiGenT and LUCES.

**Weaknesses:**

1. The use of an image-to-video diffusion model for sequence generation introduces significant computational overhead, which may limit its use in real-time or resource-constrained applications.
2. The current evaluation is restricted to object-centric normal estimation, and generalizing the approach to complex scene-centric (indoor/outdoor) settings remains a key direction for future work.

**Questions:**

See Weakness

---

> ### Author Response · Authors · 2025-11-21
> **Response to Reviewer otsa**
>
> **Q1: Significant computational overhead.**
>
> We report RoSE’s inference speed for single-image normal estimation and compare it with existing methods (see **Tab. 7** in the appendix and below). Although RoSE is slower than methods such as NiRNE, it remains faster than Neural LightRig and GeoWizard. Importantly, RoSE achieves the top two performances across all benchmarks and ranks first in average accuracy in normal estimation.  Moreover, our method is well-suited for applications that do not require real-time performance, such as serving as an auxiliary signal for 3D generation or offline normal map estimation.
>
> We have included this discussion in **Sec.D** in the appendix with the revision shown below:
>
> > _**Testing details.** … All testing processes are performed on a single RTX A6000 Ada GPU. The total runtime includes the video diffusion model inference with 25 denoising steps and the shading to normal computation, the latter adding only a negligible cost of 0.045 seconds per object. For completeness, we also report the inference time of other methods for reference._
> >
>
> **Table 7**: Average inference time of monocular normal estimation methods per image (in seconds).
>
> | **Method** | **GeoWizard** | **DSINE** | **StableNormal** | **Lotus-G** | **Lotus-D** | **Neural LightRig** | **NiRNE** | **Ours** |
> | --- | --- | --- | --- | --- | --- | --- | --- | --- |
> | Time | 101.11 | 0.83 | 1.52 | 0.61 | 0.59 | 93.73 | 0.31 | 10.57 |
>
> **Q2: Generalising to a complex scene-centric setting.**
>
> Generalising to complex scene-centric settings is indeed an important direction for future work. The primary focus of this paper is monocular object-level normal estimation, where we introduce a state-of-the-art method that first predicts a shading sequence and then estimates surface normals, together with a dataset spanning diverse shapes, materials, and lights.

---

### Official Review · Reviewer_hBMD · 2025-11-01

**Soundness:** 3
**Presentation:** 2
**Contribution:** 3
**Rating:** 6
**Confidence:** 3

**Summary:**

This paper addresses the task of estimating surface normal maps from a single RGB image. As I understand it, the motivation stems from the observation that prior single-image normal estimators often produce results that appear color-correct but fail to align with true surface geometry. The so-called “3D misalignment” problem in the paper. The idea of this paper is that, instead of directly predicting normal maps, the authors propose leveraging shading information, which, as shown in Figure 3, is more sensitive to surface detail (assuming Figure 3 is accurate). This provides both intuition and validation for their approach.

Building on this idea, the authors design a new normal estimation framework that takes advantage of a pretrained video diffusion model. Rather than training the model to predict normals directly, they fine-tune the video diffusion network to generate a sequence of shading images. These predicted shading images are then used to recover the final normal map via a solver.

Experimental results demonstrate that this method achieves state-of-the-art performance in single-view normal map estimation.

In summary, I find the motivation and design of this paper reasonable and well-aligned. I personally like the core idea, especially if there are no prior works that attempt to estimate normal maps from single RGB images through shading prediction (In the scope of neural network normal prediction from monocular images)

As for weaknesses, the paper could be improved by including more visualization materials to help readers better understand the advantages and effectiveness of the proposed approach. Details are provided below.

**Strengths:**

- I appreciate the motivation and core observation of this paper, as well as how the method design naturally follows from the motivation. Specifically, I like the logical flow, identifying an underlying mechanism that may cause problems, analyzing it, and addressing it through targeted design choices. The approach feels somewhat “old school,” but I personally find it appealing.

- The experimental results are good and convincing.

**Weaknesses:**

- The figures in the paper are of low resolution. Many images appear blurry and show visible JPEG compression artifacts when zoomed in, making it difficult to discern differences compared to the baseline methods.

- As I mentioned in the paper summary, I think the authors could include more visualization results to help readers better appreciate the framework. For example, in the prediction pipeline, the authors use a video diffusion model to predict sequences of shading images. Could the authors provide video visualizations of these sequences to show how they look?

**Questions:**

Although I like the main idea, I am curious whether Figure 3 is truly accurate. According to Equation 1, the shading map is a linear projection (the dot product between the normal vector and the light direction). As far as I understand, if the normal map itself has “3D misalignment”, as the paper describes, where if neighboring values are similar, then its linearly projected shading map should also show similar misalignment, since the neighboring values would remain correlated. Could the authors clarify this point further?

In fact, I suspect that the main reason why learning shading images works better is because shading images are more natural than normal maps. That is, they are more consistent with the distribution or prior learned by the pretrained diffusion model. I would appreciate if the authors could help address this hypothesis.

---

> ### Author Response · Authors · 2025-11-21
> **Response to Reviewer hBMD**
>
> **Q1: Low-resolution figures.**
>
> We have updated the low-resolution figures in the revised version to enable clearer zoom-in comparison.
>
> **Q2: More visualization results about the shading sequence.**
>
> We have added qualitative comparisons of the generated shading sequences in **Fig. 15** to **Fig. 27** in the appendix. We have also uploaded the corresponding videos to the anonymous link (file link: shading_videos) in the appendix.
>
> **Q3: The normal map and shading sequence share similar misalignment due to the linearity.**
>
> Our shading computation is **not a simple linear dot product** between the normal and the light direction. It further applies a max(.,0) operation (i.e., negative-clamping) to the dot product (see **Eq. 1**), thereby introducing nonlinearity to the shading sequence. Under this formulation, if the predicted normal contains misalignment, its effect on the shading sequence doesn’t necessarily follow a linear relationship.
>
> From a physical perspective, this nonlinearity is meaningful because the max(.,0) operation models “attached shadows” [A, B]. Attached shadows reveal underlying geometric information through brightness variation, especially along the boundary between illuminated and shadowed regions. This explains why the shading sequence contains rich geometric cues and is sensitive to geometric variations.
>
> We have revised the **Sec. 3.1** to explain more about the max(.,0) operation.
>
> > *… max(.,.) is the non-linear maximum operator… Shading removes the effects of surface reflectance and occlusion-induced cast shadows while preserving the geometry information and the attached shadow.*
> >
>
> [A]  Karnieli, Asaf, Ohad Fried, and Yacov Hel-Or. "Deepshadow: Neural shape from shadow." European Conference on Computer Vision. 2022
>
> [B] Xiaoyao Wei, Zongrui Li, Binjie Ding, Boxin Shi, Xudong Jiang, Gang Pan, Yanlong Cao, and Qian Zheng. Revisiting supervised learning-based photometric stereo networks. IEEE TPAMI, 2025.
>
> **Q4: The predicted shading sequences are more consistent with the diffusion model’s priors than the normal maps, which leads to a better performance.**
>
> We appreciate this insightful opinion. The latent distribution of shading maps, after being encoded by the image diffusion model VAE, appears closer to that of RGB images than to that of normal maps. This is validated by visualising the downsampled latent features of normal maps, rendered RGB images, and shading maps from 1000 randomly selected objects in the MultiShade dataset using t-SNE, as shown in **Fig. 31** in **Sec. I.1**. Although shading maps exhibit a distribution closer to that of RGB images, it is also possible that diffusion models, due to their large-scale pretraining and strong generalisation capability, can still adapt well to the distribution of normal maps, as demonstrated by methods such as Lotus.
>
> The explanation in our paper is grounded in a physical perspective: a multi-light shading sequence is sensitive to geometric variations and encodes rich geometric information through its light variations. When the video diffusion model learns to capture this information and predicts the accurate shading sequence, the normals inferred from this sequence recover more 3D-aligned geometric details, resulting in higher estimation accuracy.
>
> More details about the analysis of latent distribution can be found in **Sec. I.1** in the appendix.

---

### Author Response · Authors · 2025-11-21
**Response to All**

We sincerely thank all reviewers for their thoughtful feedback and constructive suggestions. We are encouraged by the uniformly positive assessments, with reviewers highlighting the “well-aligned design and motivation” (hBMD), the “novelty” of our work (otsa, uAAj), the “interesting technical contributions” (81qY, fSqW), and the “new perspective it brings to normal reconstruction” (fSqW). Below, we provide detailed responses to the raised questions.

We have also uploaded the revised main paper and appendix **as separate files**, with all updated content marked in blue. We will continue refining the main paper and the appendix throughout the rebuttal period.

---

### Author Response · Authors · 2025-11-30
**Summary of Paper Contributions, Reviewer Feedback, and Author Response**

We sincerely appreciate the reviewers for their time and constructive feedback, and we would like to express our gratitude to the PCs, SACs, and ACs for their timely support in addressing the unexpected incident. To facilitate the committee’s work during the upcoming post-rebuttal period, we summarise the paper’s contributions, the reviewers’ feedback, and our rebuttal responses as follows:

1. **Paper’s contributions:** We introduce a new paradigm for monocular normal estimation by leveraging a video diffusion model to predict a multi-light shading sequence, from which the normal map is subsequently derived through an ordinary least squares solver. The proposed method achieves the best average normal estimation accuracy across multiple benchmark datasets. In addition, we contribute a new dataset (MultiShade) designed specifically for this task.
2. **Reviewers’ initial responses:** We are grateful that the reviewers recognised the novelty, effectiveness, and technical contributions of our method, and provided positive initial assessments (we received 6,8,6,6,6, with an average score of 6.4).
3. **Rebuttal phase:** We provided detailed and comprehensive responses addressing all reviewers’ questions, including:

    a. additional clarifications on the technical details (hBMD: Q3; otsa: Q1, Q2; uAAj: Q1, Q3; fSqW: Q1, Q4, Q5, Q7, Q9);

    b. additional analyses, ablation studies, and comparative evaluations (hBMD: Q2, Q4; uAAj: Q2, Q4; 81qY: Q1, Q2, Q3, Q4; fSqW: Q2, Q3, Q6, Q8, Q10);

    c. minor fixes (hBMD: Q1).

    These additions strengthened the presentation and improved the clarity and completeness of the paper, which have been included in the current revision. We particularly highlight and summarise them in our responses. Notably, the paper's findings and contributions remain solid and well supported by the expanded analyses.

4. **Reviewers’ responses before the incident:** Reviewer 81qY confirmed that the rating (rate:8, confidence:5) remained unchanged after reading the rebuttal. Details are available in the corresponding discussion thread.

We appreciate the reviewers’ efforts throughout the review process. We further thank the committee in advance for the additional effort required during the upcoming post-rebuttal period, as well as the care and diligence that will be involved in making the final assessment. Finally, we hope that our method can inspire further research and offer new perspectives to the community.

Best Regards,

Authors of Paper 9653

---

### Meta-Review · Area_Chair_43zd · 2025-12-16

**Summary:**

This paper presents a novel approach to address the "3D misalignment" problem in monocular normal estimation. It leverages a video diffusion model to predict a multi-light shading sequence, diverging from traditional methods that directly predict normal maps. A normal least squares solver is then employed to derive the final normal estimation. Based on the reviewers's comment, the idea is novel, the experimental results are sound and convincing. All reviewers expressed a strong inclination to accept the paper, giving it high scores (6, 8, 6, 6).

**Reviewer Concerns:**

Concerns that are addressed:

Reviewer hBMD: Concerns include low resolution figure and lack of visualization are well addressed by the author. The two questions raised by the reviewer is also well answered.

Reviewer otsa:
1. Concerns about the computational overhead is clarified. The proposed method is time consuming but still within a reasonable budget.
2. Generalization to complex scene as a future direction is acknowledged by the author.

Reviewer uAAj: Concerns about the training details and ablation studies are well addressed by revising the paper and providing new ablation studies. Questions about inference speed and performance for transparent or translucent objects are also answered reasonable.

Reviewer 81qY:  Concerns about the quality of shading sequence, discussions about failure cases and more visualization results are all addressed by the author.

Reviewer fSqW: Concerns about similarity to NormalCrafter is well addressed. The authors also provide detailed answer to all other questions and concerns raised by fSqW. The answers are reasonable.

**Reviewer Scores:**

I believe reviewer hBMD, otsa and uAAj would raise the score a bit to 7 as I feel the authors have addressed all their concerns. Reviewer 81qY has replied to the discussion and maintain the score 8. The answer to Reviewer fSqW's question are reasonable and I feel the score may maintain to be 6.

---

### Decision · Program_Chairs · 2026-01-26

Accept (Oral)